# Small-molecule inhibition of Lats kinases may promote Yap-dependent proliferation in postmitotic mammalian tissues

Nathaniel Kastan[1,2,14], Ksenia Gnedeva [3,14✉], Theresa Alisch[1,2], Aleksandra A. Petelski[1,2,12], David J. Huggins[4,5], Jeanne Chiaravalli[6,13], Alla Aharanov[7], Avraham Shakked[7], Eldad Tzahor [7], Aaron Nagiel [8,9,10], Neil Segil [3,11] & A. J. Hudspeth[1,2]

Hippo signaling is an evolutionarily conserved pathway that restricts growth and regeneration predominantly by suppressing the activity of the transcriptional coactivator Yap. Using a high-throughput phenotypic screen, we identified a potent and non-toxic activator of Yap. In vitro kinase assays show that the compound acts as an ATP-competitive inhibitor of Lats kinases—the core enzymes in Hippo signaling. The substance prevents Yap phosphorylation and induces proliferation of supporting cells in the murine inner ear, murine cardiomyocytes, and human Müller glia in retinal organoids. RNA sequencing indicates that the inhibitor reversibly activates the expression of transcriptional Yap targets: upon withdrawal, a subset of supporting-cell progeny exits the cell cycle and upregulates genes characteristic of sensory hair cells. Our results suggest that the pharmacological inhibition of Lats kinases may promote initial stages of the proliferative regeneration of hair cells, a process thought to be permanently suppressed in the adult mammalian inner ear.

[1] Howard Hughes Medical Institute, The Rockefeller University, New York, NY, USA. [2] Laboratory of Sensory Neuroscience, The Rockefeller University, New York, NY, USA. [3] Tina and Rick Caruso Department of Otolaryngology—Head and Neck Surgery, University of Southern California, Los Angles, CA, USA. [4] Tri-Institutional Therapeutics Discovery Institute, New York, NY, USA. [5] Department of Physiology and Biophysics, Weill Cornell Medical College of Cornell University, New York, NY, USA. [6] High-Throughput Screening Resource Center, The Rockefeller University, New York, NY, USA. [7] Department of Molecular Cell Biology, Weizmann Institute of Science, Rehovot, Israel. [8] Department of Surgery Children's Hospital Los Angeles, Vision Center, Los Angeles, CA, USA. [9] Saban Research Institute, Children's Hospital Los Angeles, Los Angeles, CA, USA. [10] USC Roski Eye Institute, Department of Ophthalmology, Keck School of Medicine, University of Southern California, Los Angeles, CA, USA. [11] Eli and Edythe Broad CIRM Center for Regenerative Medicine and Stem Cell Research, University of Southern California, Los Angles, CA, USA. [12] Present address: Department of Bioengineering and Barnett Institute, Northeastern University, Boston, MA, USA. [13] Present address: Institut Pasteur, Paris, France. [14] These authors contributed equally: Nathaniel Kastan, Ksenia Gnedeva. ✉email: gnedeva@usc.edu

Initiated in response to injury, regeneration is a complex process that can restore the structure and function of damaged tissue. Some adult mammalian tissues retain a gradually declining regenerative capability. Regeneration occurs either by activation and amplification of resident stem cells, as in the epithelia of the skin and intestine, or through cellular dedifferentiation and proliferation, as in the liver. In other instances, such as central nervous and cardiac-muscle tissues, cells exhibit little or no potential for regeneration after injury[1–3].

One prominent example of a mammalian organ with poor regenerative capacity is the inner ear[4,5]. The auditory and vestibular sensory epithelia of all vertebrates possess only two major cell types: supporting cells, which play homeostatic and architectural roles, and mechanosensitive hair cells. In non-mammalian vertebrates such as birds, lost hair cells are replaced by residual supporting cells that transdifferentiate into new sensory receptors either directly or after undergoing division[6]. Although modest numbers of hair cells regenerate through supporting-cell transdifferentiation in the neonatal mammalian cochlea[7] and in neonatal and adult vestibular organs[8], after the first postnatal week no proliferative response occurs following damage[9,10].

In view of its fundamental roles in development, proliferation, stem-cell maintenance, and de-differentiation, Hippo signaling is an inviting target for driving regeneration[1,11]. The canonical Hippo pathway is a highly conserved signal-transduction cascade that comprises two pairs of core kinases. When activated by upstream signals, Mst1 and Mst2 phosphorylate Lats1 and Lats2; these proteins, in turn, phosphorylate the transcriptional co-activator Yap and its homolog Taz, adjusting the flux of these proteins so as to favor cytoplasmic localization. When the phosphorylation cascade is inactive, Yap flux into the nucleus is enhanced, leading to interaction with transcription factors of the Tead family and the initiation of cell division[12,13]. Yap signaling integrates a variety of information from the cellular environment, including biomechanical cues, cell density, cell polarity, metabolic challenges, and signals such as Notch and Wnt. The Hippo cascade is the conduit through which much, but not all, of this information is integrated into a decision regarding Yap activation[14].

The regenerative potential of the Hippo pathway has become abundantly clear in numerous organs, including the heart[3,15–17], retina[18], liver, and intestine[19]. We earlier demonstrated that Hippo signaling limits the size of the developing murine utricle, a vestibular sensory organ[19]. We further showed that the Yap–Tead complex is active during—and necessary for—growth and proliferative regeneration in the neonatal utricle[20] and organ of Corti[21]. In addition, genetic inactivation of Lats kinases is sufficient to drive cell-cycle reentry in the adult murine utricle and the chicken's basilar papilla[22]. These observations suggested that chemical activation of Yap signaling might engender supporting-cell proliferation in adult tissue, a key missing step in the regeneration of the mammalian inner ear.

In the present work, we characterize a small molecule identified in a high-throughput screen for Yap activators. The compound acts as an inhibitor of Lats kinases in vitro, suppresses Yap phosphorylation, induces cell proliferation in several cell lines and tissues, and promotes the initial stages of proliferative regeneration of the sensory receptors in the inner ear.

## Results

**Identification of activators of Yap signaling.** In some monolayer epithelial cultures, increased cell density leads to Hippo activation and thus retention and degradation of Yap protein, a process reminiscent of growth restriction during normal development[23–25].

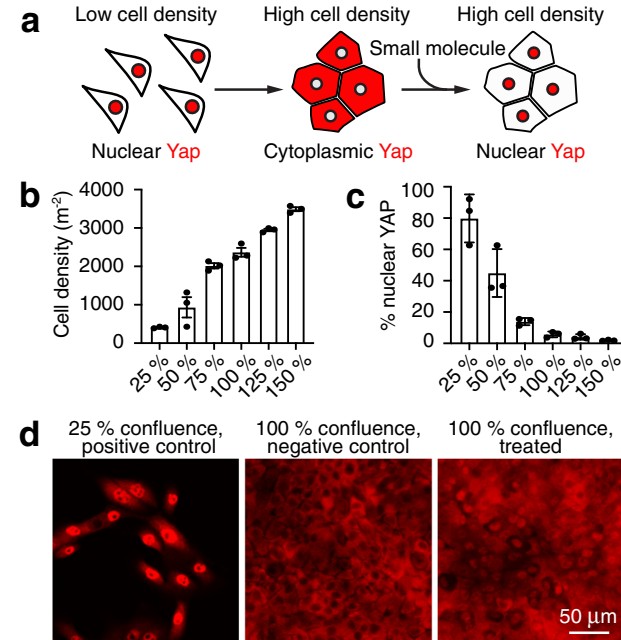

**Fig. 1 High-throughput screen for activators of Yap signaling. a** A schematic diagram demonstrates the strategy for the identification of compounds that promote the nuclear transit of Yap, thus reversing the exclusion characteristic of contact inhibition. **b** The number of MCF 10A cells rises almost linearly with their confluency ($n = 3$ for each condition, mean ± SEM). **c** Nuclear localization of Yap falls systematically with increasing confluency in MCF 10A cultures ($n = 3$ for each condition, mean ± SEM). **d** Immunolabeling of Yap (red) demonstrates nuclear localization of the protein in a positive-control (subconfluent) culture but not in a negative-control (confluent) culture. Treatment of a confluent culture with a representative hit compound promotes extensive movement of Yap into nuclei. The experiment was conducted four times with the same results. See also Supplementary Fig. 1.

To seek inhibitors of this process, we designed a high-throughput phenotypic screen for compounds that promote nuclear Yap translocation in confluent human cell cultures (Fig. 1a). After testing three lines, we chose MCF 10A mammary epithelial cells, which demonstrated a robust negative correlation between cellular confluence and the fraction of cells with nuclear Yap (Fig. 1b, c, and Supplementary Fig. 1A).

For the small-molecule screen, we seeded MCF 10A cells to achieve dense cultures in 384-well plates. A single compound was deposited in each well at a concentration of 10 μM. Every plate also included a positive control, sub-confluent cells, and negative control, densely cultured cells, both exposed to the dimethyl sulfoxide (DMSO) vehicle. After 24 h incubation, we determined the fraction of the cells with nuclear Yap and compared that value to the median negative-control value (Fig. 1d). We also scored the total number of surviving cells in each well and eliminated the compounds that decreased the number by more than one standard deviation in comparison to the dense control cultures. Nontoxic compounds that increased nuclear Yap by more than one standard deviation in comparison to the negative control were scored as hits (Supplementary Fig. 1B, C). Owing to the robustness of contact inhibition in the dense cell cultures, only six of the compounds screened met these criteria.

**Yap-dependent proliferation of supporting cells after TRULI treatment.** Using utricles isolated from mice eight to twelve weeks of age, we next tested the effects of the six compounds

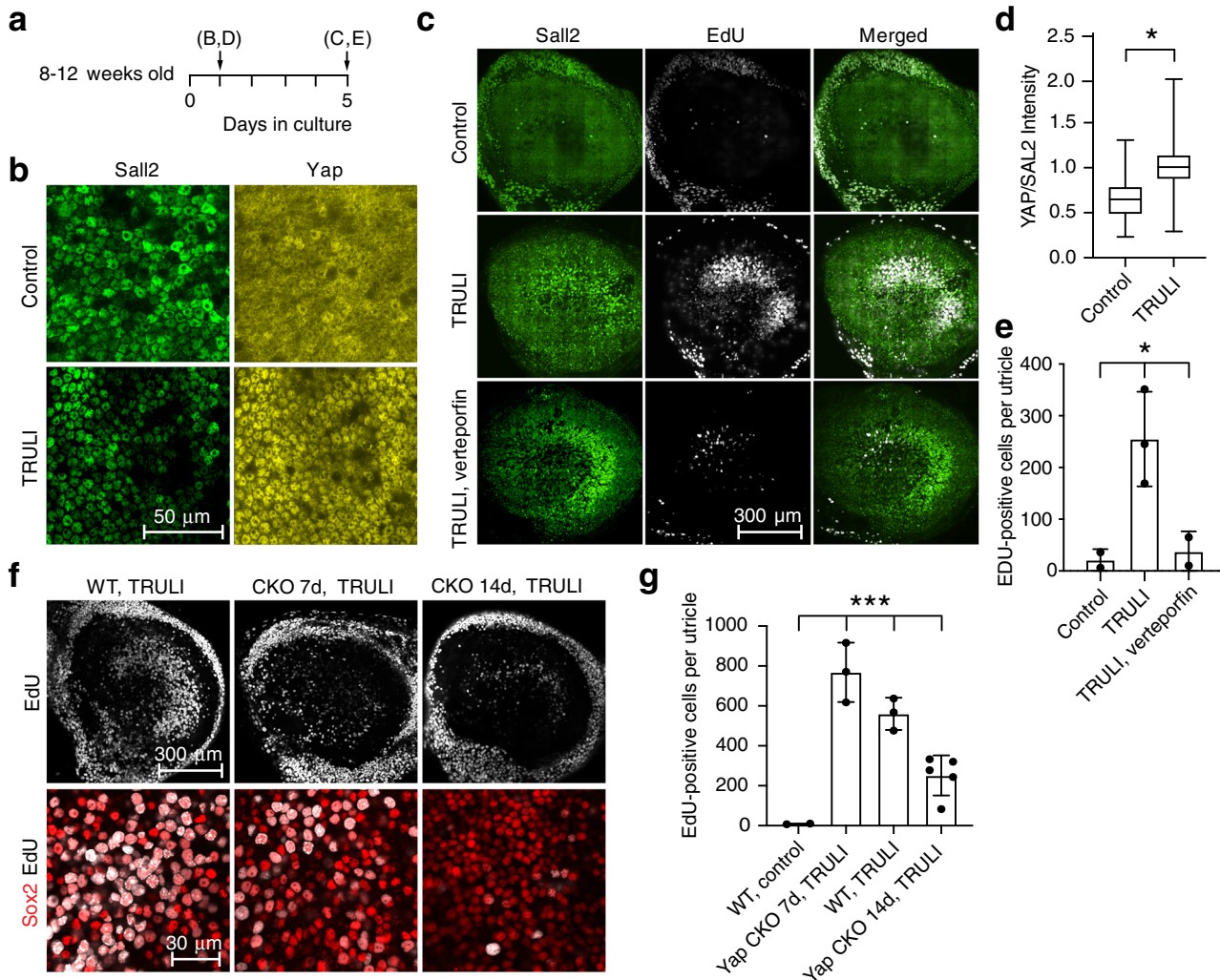

**Fig. 2 TRULI mediates activation of Yap and proliferation of supporting cells. a** A timeline depicts the pattern of the experiments shown in the indicated subsequent panels. Treatment of murine utricles with TRULI is initiated at the outset. **b** TRULI exposure drives Yap into the nuclei of utricular supporting cells. **c** In vitro exposure to TRULI for 5 days elicits proliferation of supporting cells, as measured by the incorporation of EdU. Verteporfin, an inhibitor of the Yap–Tead interaction, drastically reduces this effect. **d** In a whisker box plot, the nuclear localization of Yap is quantified as a ratio to the constitutively expressed protein Sall2 ($p = 2.683 \times 10^{-170}$, by an unpaired, two-tailed $t$-test, $n = 570$ control nuclei and 680 treated nuclei). **e** The number of EdU-positive cells per utricle increases with TRULI ($p = 0.0309$ by one-way ANOVA, control $n = 2$, TRULI $n = 3$, verteporfin plus TRULI $n = 2$, mean ± SEM). **f** Conditional deletion of Yap by tamoxifen administration to *SOX2-Cre^{ER} Yap^{fl/fl}* (Yap CKO) animals at P1, 7 or 14 days prior to explanation, decreases the number of Sox2 (red) and EdU (white) doubly positive supporting cells compared to Cre-negative *Yap^{fl/fl}* (WT) littermates. **g** Quantification of the number of EdU-positive cells in panel **f** demonstrates that decrease in supporting-cell proliferation is statistically significant ($p = 1.2798 \times 10^{-7}$ by one-way ANOVA; $n = 3$ for wild-type control and Yap CKO at 7 days; $n = 4$ for TRULI treatment; $n = 6$ for Yap CKO at 10 days and TRULI, mean ± SEM). See also Supplementary Fig. 2.

identified in the screen. Four elicited appreciable Yap nuclear translocation, but only two evoked supporting-cell proliferation. Of those two compounds, one required a concentration in excess of 100 μM and appeared toxic to hair cells. The final compound, *N*-(3-benzylthiazol-2(3H)-ylidene)-1H-pyrrolo[2,3-b]pyridine-3-carboxamide, drove robust Yap nuclear translocation after 24 h of treatment at a concentration of 10 μM and caused a striking reduction in the level of Yap phosphorylation (Figs. 2a, b, d, and Supplementary 2A). After 5 days of treatment, this substance evoked robust re-entry of adult utricular supporting cells into the cell cycle (Fig. 2c, e). For the sake of brevity—and as justified below—we term this substance "TRULI" for "The Rockefeller University Lats Inhibitor."

We also tested an inhibitor of Mst1 and Mst2 kinases[19], XMU-MP-1. Although XMU-MP-1 reduced the amount of phosphorylated Yap, it decreased the total Yap by a greater amount so that

the ratio actually increased (Supplementary Fig. 2A, B). No significant proliferation was observed after 5 days of treatment (Supplementary Fig. 2C).

We additionally assessed the effect of TRULI in the murine cochlea. Although a proliferative response was observed in Kölliker's organ, where progenitor-like cells capable of differentiating into both hair and supporting cells reside[26,27], all but the most lateral rows of supporting cells in the organ of Corti remained postmitotic after treatment with TRULI (Supplementary Fig. 3).

To test whether the proliferative effect of TRULI is exerted through Yap, we co-treated explants with TRULI and 5 μM verteporfin, an inhibitor of the interaction between Yap or Taz and Tead transcription factors[28]. Consistent with the hypothesis, co-treatment with verteporfin precluded a proliferative response (Fig. 2c, e). To confirm these results, we used *SOX2-Cre^{ER}* and

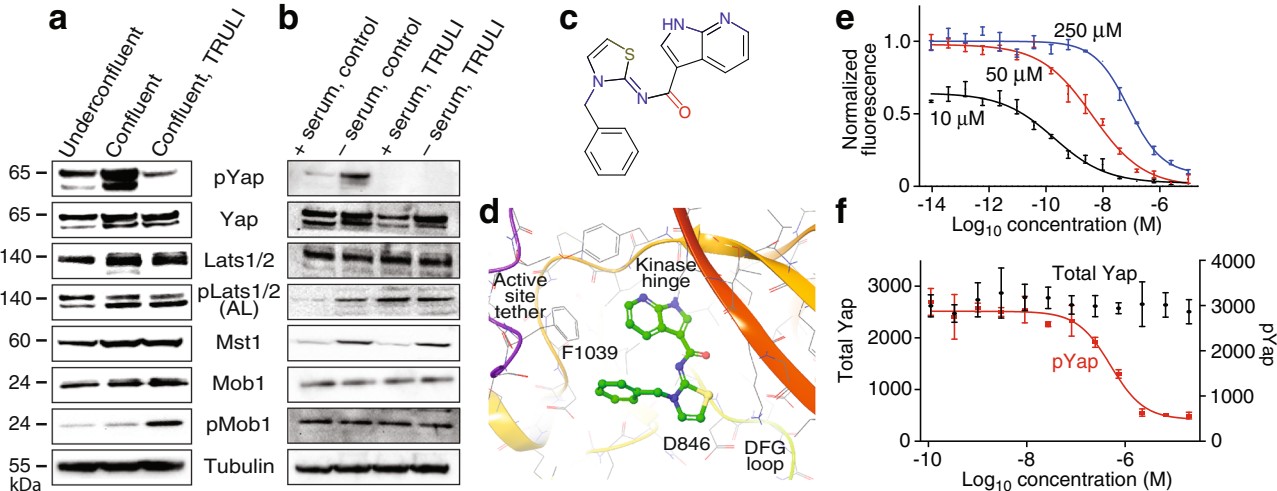

**Fig. 3 TRULI inhibits Lats activity in an in vitro kinase assay and prevents Yap phosphorylation in the cell-based assays. a** Protein immunoblotting discloses that treatment of MCF 10A cells with TRULI leaves Hippo signaling intact through the activation of Lats kinases, but that Yap phosphorylation is greatly diminished. The experiments were conducted separately three times with the same results. **b** Serum starvation of HEK293A cells in the presence of TRULI also demonstrates Lats activation but suppression of Yap phosphorylation. The experiments were run separately three times with the same results. **c** TRULI comprises a thiazolimine backbone with two substituents: a benzyl group and a 7-azaindole hinge-binding motif. **d** In a simulated structure of TRULI in the predicted ATP-binding site of Lats1, the protein is displayed as a ribbon with heavy atoms in atom-colored wire representation with gray carbons. TRULI is displayed in an atom-colored, ball-and-stick representation. Aspartate 846 is positioned to bind the thiazolimine group and phenylalanine 1039 to interact with the benzyl moiety. **e** An in vitro assay of Lats1 kinase activity shows that the $IC_{50}$ for TRULI increases with the ATP concentration, from 0.2 nM at 10 μM, to 4.3 nM at 50 μM, and to 80 nM at 250 μM, demonstrating that the compound is an ATP-competitive inhibitor. **f** A kinase assay conducted with serum-starved HEK293A cells indicates an $EC_{50}$ of 510 nM for TRULI. Starvation does not significantly deplete the total amount of Yap. In panels **e** and **f**, error bars represent SEMs, and each point represents the mean of three independent samples run in parallel. See also Supplementary Figs. 3 and 4.

$Yap^{fl/fl}$ mice to generate inducible conditional-knockout animals deficient for the protein in the sensory organs of the inner ear[29,30]. Owing to the stability of Yap protein[31], Cre-mediated recombination was induced either 7 or 14 days prior to utricular explantation and culture. In the utricles isolated from $Yap^{fl/fl}$ littermate mice lacking Cre recombinase, treatment with TRULI elicited robust proliferation of supporting cells (Fig. 2f, g). In contrast, in utricles explanted from Yap knockout animals, proliferation was significantly reduced.

**Blockage of Yap phosphorylation by TRULI in cell-based assays.** We investigated the mechanism of TRULI's inhibition in the MCF 10A cell line. After treatment of confluent cells with 10 μM of TRULI for 24 h, protein blotting revealed that the Hippo signaling cascade was intact through the phosphorylation of the activation loops of Lats1 (S909) and Lats2 (S872) (Fig. 3a). However, the phosphorylation of Yap was decreased at residue S127, a key site of Lats phosphorylation[1]. This observation suggests that TRULI is—directly or indirectly—an inhibitor of Lats kinases.

To confirm this inference we turned to HEK293A cells, which activate Lats kinases in response to serum starvation and thus inactivate Yap to prevent growth during nutrient deprivation[32,33]. We pre-treated 80% confluent HEK293A cells with 10 μM TRULI for 1 h, followed by 30 min of serum starvation. In control cultures, starvation elicited robust phosphorylation of Lats1 at S909 and the consequent phosphorylation of Yap at S127 (Fig. 3b). In cultures pre-treated with TRULI, however, serum starvation failed to evoke phosphorylation of Yap despite the activation of Lats1. Even in the treated, serum-fed conditions, phosphorylated Yap levels were below those of serum-fed control cells. Negative-feedback regulation in response to elevated Yap activity might explain why the amount of Lats1 S909 was enhanced in the treated, serum-fed condition[34]. Together, these

results suggest that TRULI interferes with the ability of Lats kinases to phosphorylate Yap.

**A potential mechanism for Lats1 and Lats2 inhibition by TRULI.** The structure of TRULI includes a 7-azaindole moiety characteristic of the hinge-binding motifs of ATP-competitive kinase inhibitors (Fig. 3c) Because there are no known crystal structures of the Lats kinases, we created a homology model from the crystal structure of the ATP pocket of similar kinase ROCK1 bound to a small-molecule inhibitor containing a 7-azaindole moiety (RCSB PDB 5KKS). A putative structure of the complex between Lats1 and TRULI was then generated by molecular docking (Fig. 3d). Because the predicted protein–ligand contact residues of Lats1 and Lats2 are almost completely conserved, the model suggests that TRULI can bind to either with similar inhibitory potencies.

To test our speculations about the mechanism of TRULI inhibition, we optimized an in vitro kinase assay using truncated forms of Lats1 (residues 589–1130) and Lats2 (residues 553–1088) that include primarily the kinase domains (Supplementary Fig. 4A–C). As a substrate, we employed the peptide STK1, which is known to be phosphorylated by these enzymes (https://www.cisbio.com/media/asset/l/s/ls-tn-lats1.pdf). Because we hypothesized TRULI was ATP-competitive, we first determined the Michaelis–Menten constants of both Lats1 and Lats2 for ATP to be near 10 μM, a concentration at which we ran the initial in vitro kinase assays (Supplementary Fig. 4A). Under these conditions, we found that TRULI inhibits both Lats1 and Lats2 with a half-maximal inhibitory concentration ($IC_{50}$) of 0.2 nM (Fig. 3e and S4D), whence the name TRULI. In support of our hypothesis, increases in ATP concentrations yielded positive shifts in the $IC_{50}$ (Fig. 3e).

To determine the potency of TRULI in living cells, we evaluated the content of total Yap and phosphorylated Yap in HEK293A cells serum-starved in the presence of various

concentrations of TRULI. The half-maximal effective concentration of the compound was $EC_{50} = 510$ nM (Fig. 3f).

To assess the selectivity of TRULI, we first compared the sequences of the putative ATP-binding residues of Lats1 and Lats2 with those of other members of the AGC kinase family (Supplementary Tables 1 and 2)[35]. To determine which of these potential off-target kinases bind the compound, we tested TRULI in a broad kinome-binding panel[36]. Of the 314 kinases tested, 34 bound TRULI more strongly than Lats1 (Supplementary Table 3). These values represent an upper bound: although only kinases bound by a small molecule might be relevant, not all such enzymes are functionally inhibited. The selectivity score, or percentage of kinases for which the inhibitor has a half-maximal concentration of binding displacement below 1 µM, was 18.1. This value compares with control values of 86.0 for the broad-spectrum kinase inhibitor staurosporine and of 18.8 for dasatinib, a clinically approved selective inhibitor of tyrosine kinases. To assess whether some of the kinases identified in both approaches were in fact functionally inhibited, we measured $IC_{50}$ values against four kinases that were high on both lists and represented by multiple family members (http://www.reactionbiology.com/webapps/site/kinaseassay.aspx?gclid=EAIaIQobChMI9K2248-V5wIVyp6zCh2HSwAKEAAYASABEgJCsPD_BwE). Some were affected significantly more strongly than others (Supplementary Table 4).

These data demonstrate that, although TRULI inhibits Lats kinases, it might also interfere with the activity of other enzymes. In the future, it will be necessary to further explore the potential off-targets of the compound, particularly in a tissue-specific context.

**Gene-expression consequences of TRULI treatment**. To further characterize the molecular effects of TRULI, we analyzed the changes in gene expression triggered by the compound after 5 days of treatment. To facilitate the sorting of supporting cells from treated utricles, we utilized *Lfng-EGFP* mice, whose supporting cells are labeled by a fluorescent reporter[37]. Principal-component analysis of RNA-sequencing data revealed that almost 60% of the variance between TRULI-treated and control samples could be explained by the first principal component and that the three samples collected under each condition clustered closely along that axis (Fig. 4a; Supplementary Data 1). Over 70% of differentially expressed genes whose expression changed by at least a factor of two were upregulated after treatment (Fig. 4b). Gene ontology analysis[38] demonstrated that the terms associated with regulation of the cell cycle were the most enriched among these up-regulated genes (Fig. 4c).

To assess the biological relevance of the changes in gene expression triggered by TRULI, we compared the FPKM values for the differentially expressed genes to those from late embryonic (E17.5) utricular supporting cells. At that stage, such cells remain highly plastic and are capable of both proliferation and differentiation into the new sensory receptors[39,40]. The expression levels for most cell cycle-related genes (gene-ontology term 0007049) that were differentially expressed in postnatal supporting cells after TRULI treatment were highly similar to those in E17.5 supporting cells (Fig. 4d). In particular, the genes specific to the S and G2/M stages of the cell cycle were significantly up-regulated by TRULI to the embryonic levels of expression (Fig. 4e, f). Consistent with the pro-survival role of Yap–Tead signaling, TRULI repressed expression of a subset of inflammatory and pro-apoptotic genes.

The direct targets of the Yap–Tead complex have been identified in other contexts, such as in mammary cancer cells[41] and, more recently, in the organ of Corti[20]. In accord with the

compound's acting as an activator of nuclear Yap signaling, over 90% of the direct downstream targets of the Yap–Tead complex among the differentially expressed genes were up-regulated in response to TRULI treatment (Fig. 4g, h). Consistent with our demonstration of a role for Yap in the development of the utricular sensory epithelium[20], most of these Yap-target genes were also highly expressed in E17.5 supporting cells.

Activation of Yap signaling by TRULI had no apparent toxic effect on utricular supporting cells: genes associated with cell death (gene-ontology term 0008219) remained unchanged after treatment (Fig. 4i).

**Supporting-cell differentiation after TRULI treatment**. Forcing postmitotic cells to re-enter the cell cycle can result in arrest at the G1-S or G2-M checkpoint transition, and consequently in programmed cell death[42]. To evaluate the long-term outcome of Yap activation on supporting cells, we treated three-week-old *Lfng-GFP* utricles with TRULI for 5 days followed by 5 days of withdrawal. RNA-sequencing analysis demonstrated that the levels of expression for the cell cycle-related genes and the direct Yap targets that were highly up-regulated upon TRULI treatment were indistinguishable between TRULI-treated and control supporting cells (Supplementary Fig. 6; Supplementary Data 2). In addition, we did not observe enrichment for pro-apoptotic genes after TRULI treatment. Instead, the up-regulated genes were associated with the gene-ontology terms related to hair-cell development and function (Fig. 5a, b). These included the genes for hair cell-specific transcription factors such as Atoh1, Pou4f3, Gfi1, and Lhx3 as well as genes encoding other important hair-cell proteins: nonmuscle myosins (Myo15, Myo7A, Myo5A), $Ca^{2+}$ sensors and buffers (Otof, Calb, S100), and transduction machinery (Tmc1, Tomt).

Because the foregoing results suggested the transdifferentiation of newly formed supporting cells, we sought newly formed hair cells by EdU incorporation. Treatment with TRULI did not affect hair-cell or supporting-cell survival, confirming that the compound is not toxic at the effective concentration (Fig. 5c, d). Although almost none of the cells expressed Pou4f3 or Myo7A, many of the EdU-positive supporting-cell nuclei migrated into the hair-cell layer. This pattern is consistent with the first stages of transdifferentiation.

To test whether supporting cells were capable of giving rise to hair cells after they had proliferated, we treated P21 utricles with TRULI and virally transfected the cultures with *Atoh1-RFP* upon drug withdrawal. Demonstrating a capacity for transdifferentiation into putative hair-cell precursors, many EdU- and Pou4f3-positive cells were observed after *Atoh1* transfection (Supplementary Fig. 6D).

**Effects of TRULI on cardiomyocytes and Müller glia**. Because the regenerative effects of activating Yap in organs such as the retina and heart have been characterized extensively[3,15,16,18], we investigated whether TRULI treatment is effective in eliciting cellular proliferation in these contexts.

The proliferative capacity of cardiomyocytes in culture and the capacity of the heart to regenerate decline sharply after birth[3]. To ascertain whether TRULI activates Yap and induces regeneration, we exposed P0 murine cardiomyocytes to TRULI for 3 days. In control cultures, protein blotting revealed high levels of phosphorylated Yap indicative of Hippo activity. TRULI treatment reduced the proportion of phosphorylated Yap and induced cardiomyocyte proliferation, as demonstrated by a significant increase in the percentage of cells positive for the markers Ki67 and phosphorylated histone H3 (Fig. 6a–e). Moreover, treated cells developed cytoskeletal rearrangements and protrusions and

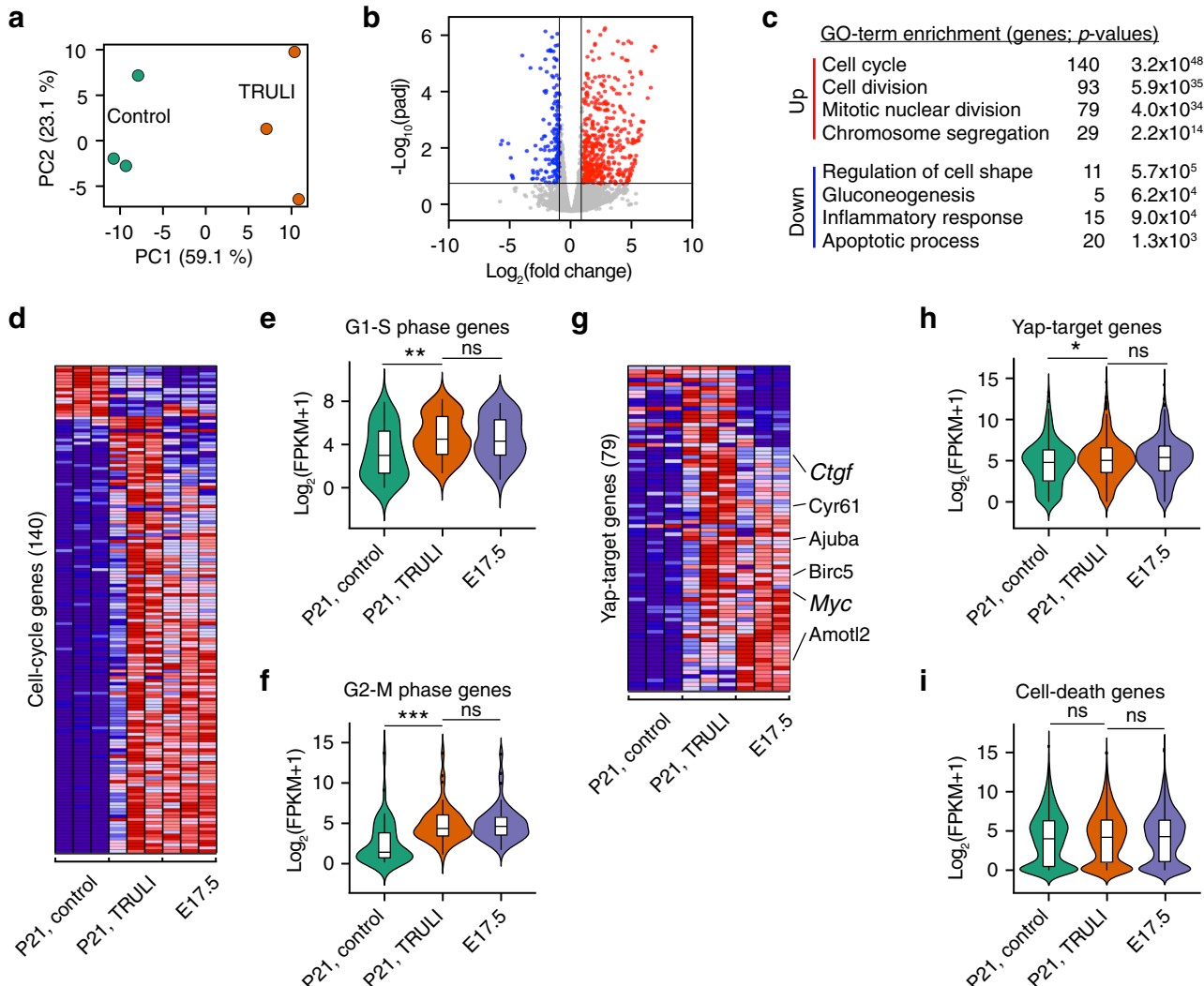

**Fig. 4 Gene-expression changes evoked by TRULI treatment in utricular supporting cells. a** For three experiments, principal-component analysis was performed on RNA-sequencing data obtained from utricular supporting cells after 5 days of treatment with TRULI or from control samples. The RNA-seq data are available through GEO (GSE148528). **b** A volcano plot visualizes the differentially expressed genes ($|\log_2(\text{fold change})| > 1$; padj < 0.05). Genes upregulated after TRULI treatment compared to control are labeled in red and those downregulated in blue. **c** Gene-ontology (GO) analysis performed with DAVID software demonstrates that the terms associated with the cell cycle are the most enriched in the genes upregulated by TRULI, whereas the terms associated with stress are more enriched in the control gene set. A full list of differentially expressed genes identified using DEseq2 is available on request. **d** For three experiments, a heatmap demonstrates the relative expression levels of 140 cell-cycle genes differentially expressed between control and TRULI-treated supporting cells (false discovery rate < 0.01) in comparison to E17.5. Highly expressed genes are shown in red and genes with relatively low levels of expression are depicted in blue. **e** Cell-cycle genes characteristic of the S phase are significantly upregulated by TRULI ($p = 0.00349$ by a two-sided Wilcoxon rank-sum test with continuity correction, $n = 42$ genes for each condition) to the levels found in E17.5 supporting cells ($p = 0.7781$ by a two-sided Wilcoxon rank sum test with continuity correction, $n = 42$ genes for each condition). Box plots indicate median (middle line), 25th, 75th percentile (box) and 5th and 95th percentile (whiskers) as well as outliers (single points). **f** Markers of the G2-M transition are likewise upregulated ($p = 3.892 \times 10^{-7}$ by a two-sided Wilcoxon rank sum test with continuity correction, $n = 52$ genes for each condition) to the levels found in E17.5 supporting cells ($p = 0.9508$ by a two-sided Wilcoxon rank sum test with continuity correction, $n = 42$ genes for each condition). Box plots indicate median (middle line), 25th, 75th percentile (box) and 5th and 95th percentile (whiskers) as well as outliers (single points). **g** For three experiments, the heatmap demonstrates the relative expression levels of 79 Yap-target genes differentially expressed between control and TRULI-treated supporting cells (false discovery rate < 0.01) in comparison to E17.5 embryonic levels. **h** The expression of Yap-target genes is upregulated in response to TRULI ($p = 0.01807$ by a two-sided Wilcoxon rank sum test with continuity correction, $n = 342$ genes for each condition) to the levels found in E17.5 supporting cells ($p = 0.3431$ by a two-sided Wilcoxon rank sum test with continuity correction, $n = 42$ genes for each condition). Box plots indicate median (middle line), 25th, 75th percentile (box) and 5th and 95th percentile (whiskers) as well as outliers (single points). **i** In contrast, the relative level of expression for genes associated with cell death remains unchanged between controls, TRULI-treated ($p = 0.06373$ by a two-sided Wilcoxon rank sum test with continuity correction, $n = 2031$ genes for each condition) and E17.5 supporting cells ($p = 0.6954$ by a two-sided Wilcoxon rank sum test with continuity correction, $n = 2031$ genes for each condition). Box plots indicate median (middle line), 25th, 75th percentile (box) and 5th and 95th percentile (whiskers) as well as outliers (single points). See also Supplementary Fig. 5.

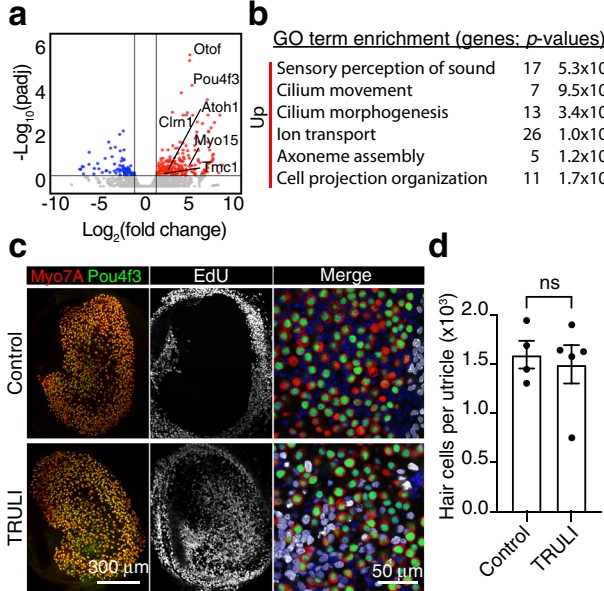

**Fig. 5 Plasticity of utricular supporting cells after TRULI treatment. a** A volcano plot visualizes the genes differentially expressed after 5 days of TRULI treatment of murine utricles followed by 5 days drug withdrawal ($|\log_2$(fold change)$| > 1$; padj < 0.05). The subset of hair-cell genes most upregulated after TRULI treatment compared to control is labeled. All the RNA-seq data are available through GEO (GSE148528). **b** Gene-ontology analysis performed with DAVID software demonstrates that terms associated with hair-cell differentiation and function are the most enriched in the genes upregulated by TRULI. **c** In representative examples of wholemount utricular explants after 5 days of TRULI treatment followed by 5 days drug withdrawal, immunolabeling for Pou4f3 (green) and Myo7a (red) depicts hair cells. New sensory receptors formed by proliferation of their precursors, supporting cells, are identified by EdU-incorporation (white) and are found only after TRULI treatment. Nuclei are labeled with DAPI. **d** The total number of hair cells per utricle does not change after TRULI treatment ($p = 0.7097$ by a Student's $t$-test; $n = 4$ for controls and $n = 5$ for TRULI-treated animals, mean ± SEM).

decreased in size, phenotypes consistent with de-differentiation and proliferation, respectively (Fig. 6f, g)[43].

Yap activation may permit retinal regeneration in mammals by stimulating the proliferation of Müller glial cells, progenitors with a capacity to differentiate into photoreceptors and neurons[18,44]. To test whether TRULI induces the proliferation of Müller glia, we used retinal organoids derived from human induced pluripotent stem cells. As expected, Müller cells remained largely quiescent in control cultures. In organoids treated for 5 days with 10 μM TRULI, however, Müller cells exhibited a robust increase in proliferation (Fig. 6h–k).

## Discussion

Most experiments demonstrating the regenerative effects of Yap activation in mammalian tissues have used transgenesis, viral transfection, or indirect approaches such as activation of G protein-coupled receptors by sphingosine-1-phosphate or epinephrine. Although these studies have provided proof-of-principle results, manipulation of Yap signaling with small molecules represents a more feasible translational approach. In fact, the only other published inhibitor of the Hippo pathway, XMU-MP-1, activates Yap signaling and facilitates healing of the liver and intestinal epithelium after damage[19]. Quinolinols have also been recently found to activate Yap-Tead signaling[45], but we

have not had an opportunity to test these compounds. It is noteworthy, however, that in our hands XMU-MP-1 did not promote supporting-cell proliferation in the utricle. This result suggests that other enzymes function redundantly to Mst kinases in the sensory epithelia. These likely include Map4K kinases, which activate Lats kinases in other contexts[46]. In fact, our RNA-sequencing data revealed that Map4K2, Map4K3, and Map4K4 are all highly expressed in the adult murine utricle. To our knowledge, no enzymes are broadly redundant to Lats1 and Lats2. Inhibition of Lats kinases might therefore provide an effective means of inactivating Hippo signaling in a variety of tissues.

Re-entry into the cell cycle is only one component of regeneration; the daughter cells must also assume their correct cell fates. Our data suggest that, in addition to inducing proliferation, TRULI treatment leaves newly formed supporting cells in a plastic state. After undergoing division, supporting cells re-enter G0 upon drug withdrawal, and many cells shift their nuclei into the hair-cell layer and activate the expression of characteristic hair-cell genes; the pattern is strikingly similar to the regenerative response in non-mammalian vertebrates[6]. Because these cells can initiate transdifferentiation upon Atoh1 overexpression, combining Hippo inhibition with inhibition of Notch signaling or Atoh1 gene therapy might constitute an effective approach to restoring hair cells.

Although Yap-Tead signaling is necessary for progenitor-cell proliferation in the organ of Corti[20], in contrast to the robust response in the adult murine utricle, the vast majority of supporting cells in the neonatal cochlea remain quiescent after 5 days of TRULI treatment. A wave of transcriptional up-regulation of *Cdkn1B*, the gene encoding P27Kip1, is known to control both the timing and the pattern of cell-cycle exit in the organ of Corti[47]. This process seems to be absent from the utricle. Our RNA-sequencing data support the hypothesis that high levels of P27Kip1 constitute a parallel repressive mechanism, for the low *Cdkn1B* expression in the adult utricle is unaffected by TRULI treatment. Activation of Wnt signaling promotes proliferation in the neonatal organ of Corti, but the effect primarily involves Kölliker's organ and the lateral Hensen's cells[48–51]. This pattern resembles that observed after treatment with TRULI. To facilitate cell-cycle re-entry, the organ of Corti might require more prolonged exposure to proliferative stimuli or treatments that target multiple signaling pathways.

Although pharmacological manipulation of the Hippo pathway might prove useful in dealing with many human diseases, much remains unknown about the potential pitfalls of Lats inhibition. In the context of the Hippo pathway, Lats1 and Lats2 are considered redundant, but each paralog also has unique functions such as estrogen-mediated signaling and p53 regulation, respectively. Furthermore, Lats kinases and Yap have ancient and conserved non-Hippo functions in spindle-assembly checkpoints and cytokinesis[52]. As a putative inhibitor of Lats kinases, TRULI also offers the ability to investigate these non-Hippo functions, and it would be prudent to understand the implications of Lats inhibition on these pathways prior to therapeutic endeavors.

Another potential concern about Lats inhibition is oncogenic transformation. In this regard the effect of systemic inhibition of Mst kinases is encouraging[19]: even after long-term administration, no oncogenic transformation was observed in mice treated with XMU-MP-1. Moreover, the impact of Lats inhibition might be mitigated by local application in isolated fluid compartments such as the endolymph of the inner ear, vitreous humor of the retina, and pericardium of the heart. Finally, the relationship of Yap to cancer is complex. Yap activation is a marker for poor prognosis in some cancers, but the opposite is true in other instances[53]. Mutations of proteins in the Hippo pathway are

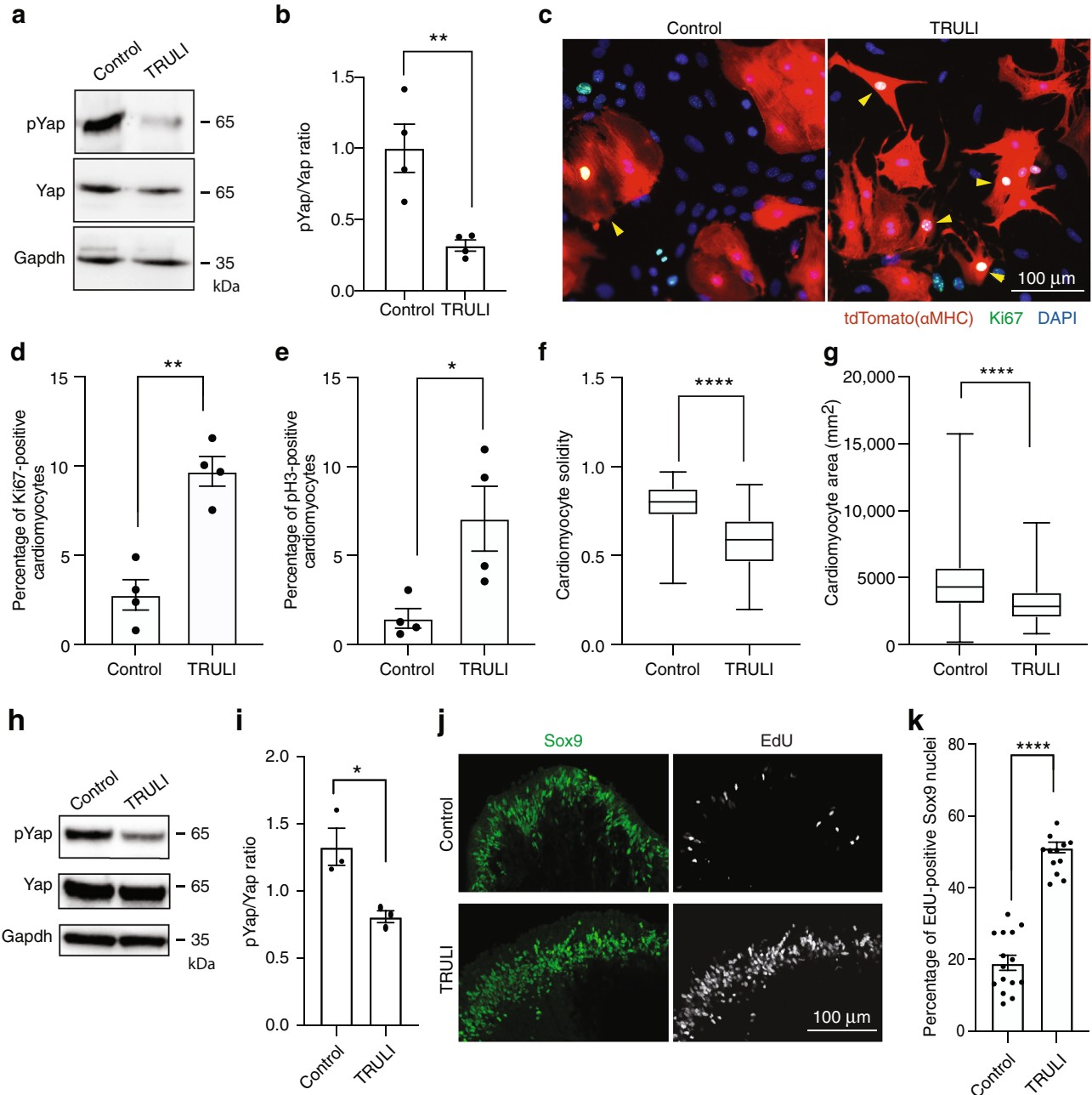

**Fig. 6 Proliferation of cardiomyocytes and Müller cells. a** A protein immunoblot indicates that TRULI decreases the phosphorylation of Yap in neonatal cardiomyocytes in vitro. **b** Quantification of the data in panel **a** shows the significance of the effect ($p = 0.008$ by an unpaired, two-tailed $t$-test, $n = 4$ independent biological samples, mean ± SEM). **c** Immunofluorescent labeling discloses that 3 days of TRULI treatment elevates the cell-cycle marker Ki67 (arrowheads) in cardiomyocytes. Scale bars, 100 μm. **d** Quantification again reveals significant effects for Ki67 ($p = 0.001$ by an unpaired, two-tailed $t$-test, $n = 4$, mean ± SEM). **e** A marker of mitotic initiation, pH3, is similarly elevated by treatment ($p = 0.026$ by an unpaired, two-tailed $t$-test, $n = 4$, mean ± SEM). **f** The cardiomyocyte solidity, an index of cellular shape, decreases significantly after treatment, an effect consistent with de-differentiation ($p = 1.09 \times 10^{-40}$, by an unpaired, two-tailed $t$-test, $n = 232$ control cells and 197 TRULI-treated cells, mean ± SEM). **g** TRULI treatment reduces the areas of cardiomyocytes, another sign of mitosis ($p = 4.09 \times 10^{-13}$ by an unpaired, two-tailed $t$-test, $n = 231$ control cells and 197 TRULI-treated cells, mean ± SEM). **h** A protein immunoblot indicates that 24 h of treatment with TRULI decreases the phosphorylation of Yap in Sox9-marked Müller cells of human retinal organoids. **i** A bar graph documents the effect observed in panel **h** ($p = 0.024$ by an unpaired, two-tailed $t$-test, $n = 3$, mean ± SEM). **j** After 5 days of TRULI treatment, EdU labeling shows a substantial increase in cellular proliferation. **k** A bar graph quantifies the result of panel **j** in three experiments ($p = 2.35 \times 10^{-13}$ by an unpaired, two-tailed $t$-test, $n = 15$ control organoids and 16 TRULI-treated organoids, mean ± SEM).

rarely causative of cancer, but are instead secondary to the molecular changes associated with tumorigenesis[11]. In fact, activation of Yap through deletion of Lats kinases in certain syngenic tumors *decreases* their progression and metastasis in immune-competent mice by augmenting the immune response[54].

Although our data from in vitro kinase assays show that TRULI is a potent inhibitor of Lats kinases, a definitive demonstration will require crystallographic confirmation of the compound's binding site. We also cannot exclude the possibility that the measured effects on Yap phosphorylation and cell

proliferation are mediated through additional mechanisms. Nevertheless, the substance is a useful tool for exploring the role of Hippo signaling in a variety of biological contexts. As expected on the basis of prior work, Yap activation with TRULI induces the proliferation of murine neonatal cardiomyocytes in primary culture and of human Müller glia in retinal organoids. Moreover, the sensory epithelia of the inner ear now join the list of tissues amenable to Yap-mediated regeneration. In view of the ubiquity of these effects, it is plausible that drugs related to this thiazolimine class will prove useful in therapeutic contexts.

## Methods

**Animal care and strains.** Experiments were conducted in accordance with the policies of the Institutional Animal Care and Use Committees of The Rockefeller University, the University of Southern California, and the Weizmann Institute of Science. All procedures were conducted in accordance with the Public Health Service Policy and the Guide for the Care and Use of Laboratory Animals. Mice were maintained in the Comparative Bioscience Center at The Rockefeller University, an AAALAC-accredited facility, under specific-pathogen-free conditions in individually ventilated isolator caging systems with 440 cm$^2$ of floor space (Thoren Caging Systems, Inc.; Hazleton, PA). Animals were housed on irradiated bedding (Anderson Bed O Cob) and received ad libitum irradiated diet (5053 PicoLab Rodent Diet 20) and hyperchlorinated water. Cages were changed weekly in laminar-airflow workstations. Temperatures were maintained as recommended by the Guide for the Care and Use of Laboratory Animals (Eighth Edition) as close as possible to the species-specific thermoneutral zone (22 ± 2 °C) and the humidity was held at at 30–70%. Housing rooms were maintained on a 12 h:12 h light–dark cycle with 10–15 changes of room air per hour.

Swiss Webster mice were obtained from Charles River Laboratories. *Sox2-CreER*, *αMHC-Cre*, and *ROSA26-tdTomato* mice were obtained from the Jackson laboratory. *Yap$^{fl/fl}$* [29] were provided by Dr. Martin, Baylor College of Medicine.

Cardiac muscle cells were lineage-traced with tdTomato-fluorescent protein by intercrossing αMHC-cre mice[55], which exhibit highly efficiency recombination in cardiomyocytes, with ROSA26-tdTomato mice[56] that require Cre-mediated recombination for expression. Both lines were maintained on a C57BL/6 background.

**Chemical libraries.** A total of 140,238 different compounds from a library at The Rockefeller University High-Throughput and Spectroscopy Resource Center were chosen by a quantitative estimate of drug-likeness score[57]. The compounds originated in the following commercially available libraries: ChemDiv (San Diego, CA), Enamine (Monmouth Junction, NJ), BioFocus (Charles River, Wilmington, MA), ChemBridge (San Diego, CA), Specs (Zoetermeer, The Netherlands), Life Chemicals (Niagara, Canada), and AMRI (Albany, NY). Compound stocks were stored in 384-well polypropylene plates at a concentration of 5 mM in DMSO at −30 °C.

**High-throughput screen for small-molecule activators of nuclear Yap.** Three human epithelial cell lines were tested for the screen: one embryonic kidney line, (HEK293T, ATCC CRL-11268) and two mammary-gland lines (MDA-MB-231, ATCC HTB-26 and MCF 10A, ATCC CRL-10317). Of the three lines, only MCF 10A cells displayed a reproducible correlation between the number of cells seeded and the number of adherent cells after 24 h in culture and thus were selected for the screen.

MCF 10A cells were cultured in a medium comprising DMEM/F12, 5% horse serum (Invitrogen 16050-122), 20 ng/L epidermal growth factor (Millipore GF155), 0.5 mg/L hydrocortisone (Sigma H-0888), 100 ng/L cholera toxin (Sigma C-8052), 10 mg/L insulin (Sigma I-1882), and antibiotic–antimycotic solution. Cell cultures reached 100% confluence at an approximate density of 3 × 10$^5$ cells per square millimeter.

For the small-molecule screen, chemical-library plates were thawed at room temperature and 0.1 μL of each compound was placed in a well of a 384-well assay plate (Greiner Bio-One) containing 10 μL of MCF 10A culture medium (PerkinElmer Janus with Nanohead). MCF 10A cells were then plated in 40 μL of MCF 10A culture medium to achieve a final concentration of 10 μM for each compound and 0.25% (vol/vol; 35 mM) DMSO. Negative (100% confluent) and positive (25% confluent) control wells included an identical concentration of the DMSO vehicle.

After 24 h incubation, the cells were fixed, washed thrice in PBS, and immunolabeled for Yap. The nuclei were stained with 3 mM 4,6-diamidino-2-phenylindole dihydrochloride (DAPI). The plates were imaged automatically with a ×10 objective lens on the ImageXpress XLS wide-field Micro reader (Molecular Devices, Sunnyvale, CA) with MetaXpress software version 4.1 (Molecular Devices). Using an empty plate to avoid out-of-focus images, we configured a laser-based autofocus routine for automatic well-bottom detection. We imaged DAPI and Yap conjugated to Alexa Fluor 488 with filter cubes for respectively DAPI (excitation, 350–400 nm; emission 415–480 nm) and FITC (excitation, 460–505

nm; emission, 510–565 nm). The acquisition system was configured to image one site per well. After the fluorescence of DAPI had been used to set the image-based autofocus, the exposure time for each fluorophore was determined with a negative-control well in which the fluorescence signal was set to 75% of the camera's maximal intensity.

An image segmentation application from MetaXpress software was used to automatically analyze the images. The number of cells was estimated by selecting an average size and intensity level above the local background in the DAPI channel.

A built-in translocation module that measured the intensity movement from one compartment to another was configured to determine the nuclear or cytoplasmic status of the Yap. These measurements were acquired first for the positive and negative control wells to determine an arbitrary threshold for the intensity ratio that characterized nuclear translocation versus cytoplasmic retardation of Yap. The same threshold was then applied to each well treated with a small molecule to identify the substances that increased the number of cells with nuclear Yap without affecting cell survival. Positive controls were not used to normalize data, but to confirm whether the cells were healthy and behaving as expected.

Images were stored in the MDCStore database and analyzed using MetaXpress software (Molecular Devices). Output data were uploaded and analyzed using the CDD Vault from Collaborative Drug Discovery (Burlingame, CA).

The compounds showing over 10% Yap nuclear translocation and over 5000 cells per field were selected and the corresponding images were checked to verify the phenotype. To determine half-maximal inhibitory concentrations, the selected compounds were re-tested in concentration–response experiments by serially diluting by half for a total of 10 dilutions to achieve assay concentrations ranging from 20 to 0.03 μM. IC$_{50}$s values were calculated by CDD software. Six compounds were confirmed in this secondary screen.

**Small-molecule Lats inhibitor.** The compound *N*-(3-benzylthiazol-2(3H)-ylidene)-1H-pyrrolo[2,3-b]pyridine-3-carboxamide (CAS number 1424635-83-5), of relative molecular mass 334.4 Da and herein termed TRULI, was originally obtained from Enamine LLC (Z730688380, Monmouth Junction, NJ). We confirmed the structure of the compound by additional synthesis and by performing nuclear magnetic resonance (NMR; Supplementary Fig. 7) and liquid chromatography–mass spectrometry (LC–MS; Supplementary Fig. 8), and used the 99 % pure material in experiments.

**Dissection and culture of inner-ear sensory epithelia.** Internal ears were dissected from mice euthanized with fluothane and placed into ice-cold Hank's balanced salt solution (HBSS; Gibco14025-092)[40].

Unless indicated otherwise, explanted cultures of the utricle and cochlea were maintained in an incubator at 37 °C in the presence of 5% CO$_2$ and 95% O$_2$. The complete growth medium comprised Dulbecco's modified Eagle medium with nutrient mixture F-12 (DMEM/F12) supplemented with 33 mM D-glucose (Sigma G8644), 19 mM NaHCO$_3$ (Sigma S8761), 15 mM HEPES (Sigma H0887), 1 mM glutamine (Sigma G8540), 5 mM nicotinamide (Sigma N0636), 40 μg/L epidermal growth factor (Sigma E9644), 20 μg/L fibroblast growth factor (Sigma F5392), insulin-transferrin-selenite solution (Sigma 11074547001), and antibiotic–antimycotic solution (Gibco 15240062).

For proliferation assays, utricles were cultured with 10 μM 5-ethynyl-2′-deoxyuridine (EdU) that was detected with click chemistry (Click-iT EdU imaging kit, Thermo C10340).

**Culture of epithelial cells.** MCF 10A cells were cultured as described above. HEK293A (Thermo R70507) cells were maintained in DMEM (Gibco 11965-092), 10% fetal bovine serum (Sigma F2442), and antibiotic–antimycotic solution. All cells were incubated at 37 °C in the presence of 5% CO$_2$ and 95% O$_2$.

**Culture of primary cardiomyocytes.** Cardiac muscle cells were lineage-traced with tdTomato-fluorescent protein by intercrossing *αMHC Cre* mice[55], which exhibit highly efficiency recombination in cardiomyocytes, with *ROSA26-tdTomato* mice[56] that require Cre-mediated recombination for expression. Both lines were maintained on a C57BL/6 background.

Neonatal primary cardiac cultures were isolated from P0 pups with a neonatal dissociation kit (Miltenyi Biotec,130-098-373) and homogenizer (gentleMACS). Cardiac cultures were seeded in gelatin-coated wells coated with 0.1% gelatin (G1393, Sigma) in DMEM/F12 medium supplemented with 1% L-glutamine, 1% sodium pyruvate, 1% nonessential amino acids, 1% penicillin–streptomycin solution, 5% horse serum, and 10% fetal bovine serum. After culture in 5% CO$_2$ for 24 h at 37 °C, the medium was replaced for an additional 72 h with bovine serum-free medium containing 0.1% (vol/vol) DMSO and, for experimental samples, 20 μM TRULI.

**Culture of human pluripotent stem cells and retinal organoids.** The WTC-11 line of induced pluripotent stem cells (Coriell Institute for Medical Research, Camden, NJ) was maintained using standard methods. Human retinal organoids were produced from these cells[58]. In three independent proliferation assays, five organoids 225–280 days of age per experimental condition were sampled in culture

after incubation for 5 days in 10 μM TRULI and 10 μM EdU. The organoids used in the immunoblot analysis were 160–178 days of age and were incubated for 24 h in 10 μM TRULI.

**Immunohistochemistry of murine inner-ear sensory epithelia.** Utricles were fixed in 4% formaldehyde (Thermo 28906) for 1 h at room temperature and then blocked for 2 h at room temperature with 3% bovine serum albumin (BSA; Jackson AB 2336646), 3% normal donkey serum (Sigma-Aldrich D9663), and 0.3% Triton X-100 (Sigma 93443), in Tris-buffered saline solution (Thermo 28358).

The primary antisera—goat anti-Sox2 (R&D AF2018, 1:1000), rabbit anti-myosin 7A (Proteus 25-6790, 1:100), mouse anti-Yap (SC-101199, 1:500), rabbit anti-Sall2 (HPA004162, 1:200), and mouse anti-Pou4f3 (SC-81980, 1:100)—were reconstituted in blocking solution and applied overnight at 4 °C.

Samples were washed with phosphate-buffered saline solution supplemented with 0.1% Tween 20 (Sigma-Aldrich), after which Alexa Fluor-labeled secondary antisera (Life Technologies) were applied in the same solution for 1 h at room temperature, all at 1:500.

Nuclei were stained with 3 mM DAPI.

**Immunofluorescence assays for cardiomyocytes.** Cells were fixed in 4% formaldehyde for 10 min with shaking at room temperature, permeabilized for 5 min with 0.5% Triton X-100 in PBS, and blocked for 1 h at room temperature with 5% BSA in PBS containing 0.1% Triton X-100. The cells were labeled overnight at 4 °C with the primary antibodies anti-Ki67 (1:200, 275R, Cell Marque) and anti-phosphorylated histone 3 (1:200, 9701, Cell Signaling). After three washes with PBS, samples were labeled for 1 h at room temperature with fluorescent secondary antibodies (Abcam) followed by 10 min of DAPI staining for nuclear visualization. After three washes in PBS, cells were imaged with a Nikon Eclipse Ti2 microscope.

**Immunofluorescence imaging of retinal organoids.** Fresh frozen sections were permeabilized and blocked in a humidified chamber for 1 h at room temperature with 3% horse serum in PBS with 0.3% Triton X-100. The slides were exposed for 2 h to primary antibodies diluted in the same solution. Anti-Sox-9 (1:100, Rb Cell Signaling 82630) was used to mark Müller cells. Slides were washed three times with PBS. Secondary antisera (Alexa-Fluor 488 and 647) diluted 1:10,000 in the serum solution were added for 1 h at room temperature followed by two washes with PBS. Labeling for 30 min at room temperature (Apply Click-iT Invitrogen Alexa Fluor 555 C10338) was followed by a wash with PBS. After incubation with DAPI (1:10,000 in PBS) for 5 min and the application of mounting solution, sections were imaged by confocal microscopy with a Zeiss LSM 700 system with a ×20/0.8 NA objective lens with a pinhole size set at the first Airy disc.

**Protein immunoblotting and quantification.** MCF 10A, HEK293 cells, retinal organoids, and utricles were lysed on ice in radioimmunoprecipitation assay buffer solution (RIPA; BP-115-5x) with protease inhibitors (Halt Protease Inhibitor Cocktail, Thermo 87786). Utricles were additionally sonicated thrice at low power for 10 s, with breaks of 20 s with the samples kept on ice between sonications. After lysates had been scraped and centrifuged at 21,130×g for 10 min at 4 °C, the supernatants were immediately subjected to electrophoresis or stored at −80 °C.

A standard immunoblotting protocol was used with the following specifications. A 4–12% bis–tris gel (Thermo NP0322) was used to resolve the proteins in 5 mg of each sample. The proteins were transferred to a nitrocellulose membrane (BioRad 1704156) and blocked for 1 h at room temperature (Rockland MB-070). After primary antibodies had been reconstituted in the same solution, the membrane was incubated overnight at 4 °C. After three 5 min washes at room temperature in tris-buffered saline solution with 0.05% Tween 20, a secondary antibody conjugated to horseradish peroxidase (Millipore, 1:10,000) was applied in the same solution for 1 h at room temperature before the activity was detected (SuperSignal West Pico PLUS, Thermo 34580). Images were acquired with an iBrightFL1000 system.

We used at a dilution of 1:1000 primary antibodies directed against Yap (sc-101199), phosphorylated Yap S127 (CST 4911), Lats1 and Lats2 (Abcam, ab70565), phospho-Lats1 S909 (CST 9157), Mst1 (CST 3682), Mob1 (CST 13730), phospho-Mob1 T35 (CST 8699), tubulin (Sigma T6793), and GAPDH (Abcam ab8245).

Cultured cardiomyocytes were lysed using RIPA supplemented with protease and phosphatase inhibitors (1:100, Sigma). Lysates were prepared and 30 μg protein of each sample was fractionated by gel electrophoresis in Tris–glycine acrylamide gels, and subsequently transferred to a PVDF membrane. Following 1 h blocking at room temperature, membranes were incubated with the following antibodies against phosphorylated Yap at S112 (1:1000, 13008, cell signaling), Yap (1:2000, NB110-58358, Novus), and Rabbit anti-GAPDH (1:8000, PLA0125, Sigma) for 1 h at room temperature. Following two washes in a Tris-buffered saline solution containing 0.1% Tween 20, membranes were incubated with horseradish peroxidase anti-rabbit or secondary antibodies (Jackson). The signal was detected by a super-signal west pico plus chemiluminescence kit (34580, Thermo-Fisher).

Immunoblots were quantified through measurement of the band intensity with Fiji[59]. The final value of each band's intensity was normalized by the sum of all the bands' intensities. Significances from Student's $t$-tests are denoted as follows: *$p <$ 0.05; **$p < 0.01$; ***$p < 0.001$; ****$p < 0.0001$. Unless otherwise indicated, error bars denote standard errors of means (SEMs).

**Imaging of Yap nuclearization and quantification of proliferation.** Confocal imaging was conducted with a confocal microscope enhanced with structured illumination (VT-iSIM, VisiTech International Ltd.).

For quantification of Yap nuclearization, for each condition, two fields of supporting cells were imaged at ×60 from two utricles. Using CellProlifer[60], we masked Sall2-positive nuclei and measured the intensity of their labeling. The same mask was then applied to the Yap channel and the intensity of labeling was measured. The ratio of the Yap to the Sall2 intensity was calculated for each condition in 680 cells after TRULI treatment and 570 cells for DMSO controls.

For quantifying the proliferation of supporting cells, we imaged utricles at ×60, then assembled the images into a composite tiling (Grid Collection Stitching, Fiji). The maximal values in z-stacks were projected to display the entire supporting-cell layer of the sensory epithelium, excluding connective tissue and the surrounding epithelium. Using CellProlifer[60], we counted the number of EdU-positive nuclei. Alternatively, EdU- and Sox-doubly positive nuclei were counted using Multi-point Tool in ImageJ.

Images acquired from neonatal cardiac cultures were analyzed using ImageJ software by thresholding for cardiomyocytes according to endogenous tdTomato fluorescence. Thresholded figures were carefully and manually separated by fine lines to de-cluster cells in case they touched, basing on original images, following by measurements of area and solidity.

Images acquired from retina organoids were analyzed with CellProfiler[60]. Masks were created for all the Sox9-positive nuclei, and the percentage of EdU- and Sox9-positive nuclei were counted.

**In vitro kinase assay.** The in vitro kinase assay (HTRF KinEASE-STK S1, CisBio 62ST1PEB) was optimized to the linear reaction range of the enzymes Lats1 (Carna 01-123) and Lats2 (Carna 01-124). Reactions were conducted with 10 μM STK1 substrate and 10 μM ATP, unless otherwise indicated. For the ATP-shift assay, ATP was also used at concentrations of 50 and 250 μM. Lats1 was employed at a concentration of 200 pg/μL and Lats2 at 50 pg/μL unless otherwise indicated. For the assay in which Michaelis–Menten constants for ATP were determined, each enzyme was at a concentration of 62.5 pg/μL; Lats1 ran for 30 min, and Lats2 ran for 20 min, all at room temperature. The DMSO concentration was maintained at 0.5% (vol/vol) throughout all experiments, and a Janus 384 MDT (PerkinElmer) equipped with a 50 nL Pintool (V&P Scientific, Inc.) was used to add the compounds dissolved in DMSO to the reaction. The enzyme, substrate, ATP, and TRULI were combined in a low-volume 384-well plate and shaken for 50 min at room temperature unless otherwise indicated. The reaction was stopped by adding the detection reagents, which were prepared at an 8:1 biotin:streptavidin ratio, and shaken for 60 min at room temperature. All reactions were conducted in triplicate and a Synergy NEO (Biotek) was used to detect the signal.

**Cellular kinase assay.** HEK293A cells were plated overnight in 96-well culture plates at a density of 50,000 cells per well in the medium described above. To start the assay, new serum-free medium with various concentrations of the compound was added to the cells, which were then incubated for 30 min at 37 °C. The DMSO concentration was kept at 0.1% (vol/vol), and all experiments were performed in triplicate.

Total Yap and phosphorylated Yap were detected according to the two-plate protocol for adherent cells (Total Yap Cellular Kit, Cisbio 64YATPEG; phosphorylated Yap [Ser127] Cellular Kit, Cisbio 64YapPEG). The signal was detected by Synergy NEO (Biotek).

**Modeling of enzyme structure.** Lats1 and Lats2 are in the AGC kinase family, which also contains kinase sub-families such as Ndr and Rock[35]. In *Homo sapiens*, Lats1 has an overall sequence identity of 51.7% with Lats2, of 42.2% with Ndr2, and of 39.9% with Rock1. Owing to the presence of the classic kinase hinge-binding azaindole motif in TRULI, we focused on the kinase domain and analyzed the ATP-binding site in order to generate a homology model. To define this site we used the 36 kinase-domain residues defined by Huang et al.[61]. We show an alignment of these ATP-binding site residues for 62 human AGC kinases (Supplementary Table 1) and the identities of these residues between Lats and the other AGC kinases (Supplementary Table 2). It is worth noting that the sequence identity between Lats1 and Lats2 in the ATP-binding site is 97.2%, with only a single conservative difference (lysine to arginine).

To generate a homology model of the kinase domain we selected a complex of Rock1 bound to an azaindole thiazole inhibitor (RCSB PDB 5KKS)[62]. This complex was selected because the sequence identity between Lats1 and Rock1 in the ATP-binding site is 72.2% and the bound inhibitor shares an azaindole group with TRULI. Although the resolution of the structure is a modest 0.33 nm (3.3 Å), the electron density within the ATP-binding site is good and allows the conformation of the ligand and the amino acid side-chains to be determined unambiguously. The protein structure was downloaded from the Protein Databank, selenomethionines were changed to methionines, and missing sidechains were added with Schrodinger's Preparation Wizard. This program was also used to fix the orientations of the asparagine, glutamine, and histidine residues as well as the protonation states of all ionizable residues. Heteroatomic species such as water molecules, buffer solvents, and ions were removed and the complex was energy

minimized with a restraint of the all-atom root-mean-square distance to 0.03 nm (0.3 Å). The homology model was generated from residues 705-1043 of Lats1 with a knowledge-based model in Schrodinger's Advanced Homology tool. Chain B of ROCK1 was used as the template and the azaindole thiazole inhibitor was retained.

The resulting homology model was then used for molecular docking with Schrodinger's Glide SP[63]. The Glide grid was generated around the azaindole thiazole inhibitor in the ATP-binding site with T845 as a rotatable group. TRULI was prepared using Schrodinger's Ligprep and docked using Schrodinger's Glide SP using default input parameters. The Glide SP docking score was −39 kJ/mol (−9.4 kcal/mol), corresponding to a binding affinity of ~130 nM.

The ligand's azaindole group is predicted to bind to the hinge region as expected, whereas the ligand's thiazolimine group contacts D846 in the DFG loop and the ligand's benzyl group contacts the C-terminal F1039, which forms the active-site tether in many AGC kinases[64].

Although the numbers assigned to the amino-acid residues of various proteins elsewhere in the text refer to the murine products, those in the paragraphs above reference the human proteins.

**RNA sequencing and data analysis**. Supporting cells were isolated by fluorescence-activated cell sorting from the utricles of *Lfng-EGFP* mice as described previously[65]. The total RNA was extracted using Quick-RNA MicroPrep kit (Zymo Research) and stored for up to 2 weeks at −80 °C. RNA samples were then processed for library preparation with QIAseq FX Single Cell RNA Library Kit (Qiagen) and the quality of the library was confirmed using a Bioanalyzer (Quick Biology Inc.). A minimum of two biological replicates was collected for each condition, and at least 20 million 150 base-paired-end reads were sequenced for each replicate. Reads were mapped to GRCm38/mm10 genome assembly using STAR[66]. Differentially expressed genes were identified by DESeq2[67]. Only the protein-coding sequences were considered for FPKM calculation. For data visualization, the ggplot2 package was used (H. Wickham, Springer New York, 2009). Principal-components analysis was performed with pcaExplorer[68] on the 1000 genes whose differential expression was most significant.

**Adenoviral gene transfer**. The AdEasy Adenoviral Vector System (Agilent) was used to create adenoviral vectors containing the full-length coding sequence of murine Atoh1 under the control of a cytomegalovirus promoter. To permit the identification of infected cells, we also included sequences for an internal ribosome entry site and for RFP. Viral particles were amplified and purified by CsCl-gradient centrifugation followed by dialysis (Viral Vector Core Facility, Sanford Burnham Prebys Medical Discovery Institute). Utricles were infected in 200 µL of culture medium with 10 µL of the virus at a titer of $10^{10}$ pfu/mL. Ad-RFP virus (Vector Biolabs) at the same titer was used as a control.

**Reporting summary**. Further information on research design is available in the Nature Research Reporting Summary linked to this article.

## Data availability
The datasets generated during the current study have been deposited to NCBI's Gene Expression Omnibus (GEO) database, accession number GSE148528. For each figure, raw data from cell counting, intensity measurements for the immunohistochemistry, and images whole Western blots that include weight markers are provided in the source data file. PDB structure of ROCK1 is available at RCSB PDB 5KKS). Source data are provided with this paper.

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

## Acknowledgements

This project was made possible by the dedicated personnel of two service facilities. At Rockefeller University's High-Throughput Screening Resource Center directed by Fraser Glickman, Carolina Adura Alcaino helped design assays for enzyme activity. The effort was supported in part by the Robertson Therapeutic Development Fund. At the Tri-Institutional Therapeutics Discovery Institute (TDI) supervised by Peter Meinke, Leigh Baxt, Stacia Kargman, Robert Myers, and Nigel Liverton assisted with assay design and Rui Liang performed chemical syntheses. TDI is a 501(c)(3) organization that receives funding from its parent institutes (The Rockefeller University, Memorial Sloan Kettering Cancer Center, and Weill Cornell Medical College), from Takeda Pharmaceutical Company, and from Mr. Lewis Sanders and other philanthropic sources. Masha Volo-godskaia, Welly Makmura, Juan Llamas, Angela Ferrario, Rosanna Calderon, and Kayla Stepanian provided technical assistance. The pipeline for RNA sequencing data quality control and alignment was modified from EndCode by Litao Tao, Talon Trecek, and Francis James. We thank the members of our research groups for comments on the manuscript. N.K. was supported by the NIGMS (T32GM007739); K.G. by the NIDCD (R21DC016984); T.A. by the Studienstiftung des deutschen Volkes; A.A., A.S., and E.T. by European Research Council (ERC AdG 788194, CardHeal), the Israel Science Foundation (ISF), and the Foundations Leducq and Minerva; A.N. by Research to Prevent Blindness and the Las Madrinas Endowment in Experimental Therapeutics for Ophthalmology; and N.S. by the Hearing Restoration Program of the Hearing Health Foundation and the NIDCD (R01DC015829 and R01DC015530). A.J.H. is an Investigator of Howard Hughes Medical Institute.

## Author contributions

Conceptualization, K.G., A.J.H., and N.K.; Methodology, J.C., D.J.H.; Formal analysis, K.G. and N.K.; Investigation: K.G., N.K., T.A., A.A.P., J.C., A.A., and A.S.; Resources, E.T., A.N., N.S., and A.J.H.; Writing—original draft, N.K.; Writing—review and editing, A.J.H., K.G., N.K., N.S.; Visualization, K.G., N.K., D.J.H., and A.J.H.; Supervision, A.J.H.; Funding acquisition, E.T., A.N., N.S., and A.J.H.

## Competing interests

K.G., N.K., A.J.H., D.H., T.A. and A.A.P. are parties to an application for patent protection of derivatives of the Lats inhibitor presented here. The remaining authors declare no competing interests.
