## [Peer Review File · Nature Communications]

REVIEWER COMMENTS

Reviewer #1 (Remarks to the Author):

This is a well-done and well-written study on Hippo signaling in the mouse inner ear and cardiomyocytes, and human Muller glia in retinal organoids. In all three organ systems, regeneration in humans is restricted/ blocked. Thus, any efforts to understanding these blocks is a significant advance in the biomedical field. When the Hippo pathway is in its active state, the transcription factors, Yap/Taz are targeted for degradation and thereby, inhibiting proliferation. When the Mst and Lats proteins are inhibited, the pathway is turned off, allowing Yap/Taz to activate transcription of G1-S and G2-M check point genes. In this collaboration, the authors mechanistically test the Lats inhibitor, TRULI, to stimulate proliferation.

This study is beneficial and of general interest to the biomedical field and I recommend this manuscript for publication if the authors can satisfy the requirements below. There are a few issues with some experiments with sample sizes and the appropriate statistical tests employed.

Major Comments:

1. The experiment in the cochlea described on page 6 (Figure S3) was done on P1 cochleas. Samples size is not noted, unlike in Fig. S2 (utricle). Such proliferative effects at this stage of development are usually only seen in the apex. What part of the cochlea were these images from? Base, mid, apex? In Fig. S3B, HCs in the TRULI condition appear to be 'blebbing', which can be a sign of toxicity (in the cochlea). Please comment on this and note how many samples exhibit this. There appears to be absolutely no proliferation in the organ of Corti itself. Is the cell cycle inhibitor, p27kip1 modified in any way?

2. On page 7, "Even under treated, serum-fed conditions, phosphor-Yap levels were below those of serum-fed control cells."

What do you mean by 'under treated'?

3. In Figure 2 (page 35),

D- The authors mention the number of cells, but not the number of organs used. Please note, N= number of experiments. The authors should sample across more than 1 utricle under both conditions. Please clarify.

E- The sample size noted is n=2 and uses a student's t-test. First, sample sizes n=2 to 3 aren't sufficient to ensure normal distribution of samples that is required for a t-test. Second, you can't use a t-test to compare across 3 samples in the way the authors show. You must use an ANOVA.

G- same problem. Can't use a t-test here. Must use an ANOVA.

In both cases E and G, the sample sizes are too small to show normal distribution. In my experience, an n=5 could demonstrate this. But normal distribution needs to be determined first.

Under the reporting summary, "A description of any assumptions or corrections, such as tests of normality and adjustment for multiple comparisons" is applicable and must be satisfied. Although significance may not change with additional samples, it is important that the correct statistical tests

are applied.

4. In consistencies in Figure 6 (page 42). In F and G, across how many experiments were these measurements taken? For example, in K, the authors clearly noted that there were 3 experiments.

D-G. Since several comparisons are being made from (supposedly) the same data, a correction factor to the p-value must be applied e.g Bonferroni correction.

5. In the legends on Figure 5 (page 41), I'm not sure why the authors only make the full list of DE genes upon request. Shouldn't this be a downloadable file from GEO?

6. I understand that the authors only used an n=2 for RNA sequencing; however, in this case secondary validation of a few selected genes by RT-qPCR (or in situ hybridization) is necessary as RNA sequencing is not sensitive enough and can throw out false positives. Especially in this case, when proliferation is affected. More cells than skew quantitative results. In this case, I recommend RT-qPCR.

Minor Comments:

7. On page 5, "Every plate also included a positive control, sub-confluent cells, and a negative control, densely cultured cells, both exposed to dimethyl sulfoxide (DMSO) at a concentration equivalent to that in which the compounds were applied."

Just succinctly refer to it as a 'DMSO vehicle control'. It is understood that you used the same concentration of DMSO.

8. TRULI is first mentioned on page 6, but its abbreviation is described very late on page 8, as the "The Rockefeller University Lats inhibitor". This should be mentioned earlier on page 6.

9. Figure S6, In the figure title/ legend please clarify that these are Utricles.

10. On page 11, "...we treated P21 utricles with TRULI and virally transfected the cultures with Atoh1-RFP upon drug withdrawal." The way this reads, Atoh1-RFP is a reporter for Atoh1. If I am not mistaken, the Methods section describes it as an 'Atoh1-RFP expression construct'. If the authors intended only to observe the capacity for HC differentiation upon TRULI treatment, please explain the rationale for using an expression construct rather than a reporter construct.

11. On page 13, second paragraph, 'cochlea' is mis-spelled as 'cocTiktaalikhlea'.

Reviewer #2 (Remarks to the Author):

The authors report the identification of a small molecule that inhibit the LATS kinases, which are key tumor suppressors in the Hippo pathway. This signaling pathway is highly conserved and an essential regulator of development and disease. Pharma and academia are pursuing efforts to target it for therapeutic benefit, predominantly in the context of cancer (mostly YAP, TAZ, TEAD inhibitors) and regenerative medicine. There has been some progress on the latter front, but this has centred on

the upstream Hippo kinase (human MST1 and MST2). This study is original by virtue of the fact that it centres on the key downstream kinases, LATS1 and 2, which are more potent (based on genetic loss of function studies in flies and more recently mammals).

The authors use an elegant cell density screen to identify a handful of compounds that reinstate nuclear YAP under confluent conditions and focus on TRULI. They subsequently show that this is an ATP-competitive inhibitor of LATS1 and 2. They convincingly show using RNA-seq that YAP hyperactivation is a major downstream effect of the drug. They also show that it has promising regenerative capacity in different tissues, most notably the utricle. TRULI induces cell cycle re-entry but also plasticity and this enable newly generated cells to adopt specific fates, which is important for tissue repair. TRULI should be a useful tool compound for Hippo studies and is a promising candidate to be developed for potential use in regenerative medicine, especially because it should be more potent than MST1/2 inhibitors for YAP activation. Its study will also help to determine whether short term Hippo pathway disruption is safe and does not induce unwanted effects, like YAAP-driven oncogenesis.

This is a high-quality study and should be published in Nature Comms. I have only one suggested experiment (which is essential) and minor comments in order to improve the paper.

The studies in the utricle and retinal organoids report that TRULI induces proliferation, based on EdU positivity. The authors have not actually shown that cell number increases in response to TRULI, just that DNA replication is stimulated. Cell number and mitosis (e.g. with phospho-histone H3) should be assessed in order to more rigorously test this key assertion. I note that they have assessed mitosis in cardiomyocytes but cell number should also be tested here and mitosis and cell number should be assessed in the other settings.

Some comments on the text:

- Despite the fact that the Hippo pathway is commonly described as being in active or inactive states and YAP and TAZ to be sequestered in the cytoplasm and degraded in response to Hippo, live imaging studies in cultured cells and flies have now shown these assumptions to be largely wrong. In fact, the authors' own data (Figure 1A) reveals the limitations of this broad statement, with respect to Hippo signaling and YAP degradation. These assumptions should no longer be perpetuated in the literature.

- All statistical information (error tests, n numbers, etc.) should be provided in the figure legends. This is not always provided.

- Legends for Figures 2, 4, 5 and 6 need more descriptive titles. Non-specialists won't know what "supporting cells" are. Add the organ name.

Reviewer #3 (Remarks to the Author):

Comments to the authors:

Herein, Kastan et al. reported the discovery of non-toxic, reversible LATS1/2 kinase inhibitor namely TRULI through high throughput phenotypic screen, which is a very important and inviting topic in regeneration medicine. TRULI promoted nuclear YAP translocation and evoked supporting-cell

proliferation. Direct binding to LATS1/2, ATP competitive MOA and global kinome selectivity were demonstrated in in vitro kinase assays. The upregulation of cell cycle related genes and YAP target genes triggered by compound was evaluated by RNA-Seq. Additionally, TRULI could induce supporting-cell differentiation and stimulate the proliferation of cardiomyocytes and Müller glial cells. Overall, it's an interesting and solid study. However, several issues need to be addressed to increase the confidence in the specificity.

Major points:

1. TRULI shows LATS1/2 inhibition with IC50 value of 0.2 nM while cellular potency differs by three orders of magnitude (EC50 = 510 nM). Any explanation for this discrepancy? Some phenotypic experiments were conducted in 10 µM, which is significantly higher than in vitro biochemical potency. Is that due to poor cell permeability or mM-level cellular ATP concentration, which positively shifted IC50 of ATP-competitive TRULI in cells (indicated from Figure 3E)?

2. Also concerning the selectivity issue, it would be better to show LATS1/2 KO result or introduce resistant mutation based on docking model to make LATS1/2 refractory to inhibitor to demonstrate the observed phenotype is due to direct LATS1/2 inhibition. Alternatively, or in addition, it would be good to show basic structure-activity relationships for this compound. This would help address the question of whether the activity is primarily due to LATS1/2 inhibition or the consequence of polypharmacology. So showing correlation between biochemical LATS1/2 inhibition and cellular activity would lend support to the notion that the activity is indeed 'on-target'.

3. The authors tested kinome selectivity (314 kinases) via in vitro binding assays. 34 kinases bound more strongly than LATS1/2, although it's acceptable considering it's an unoptimized hit. Would be interesting to see cell-based selectivity on this. ActivX Biosciences provide KiNativ™ that could be considered for in situ kinase profiling and on-target validation in cell models. Also, many kinases such as MAP4K would have crosstalk with Hippo signaling cascade (DOI: 10.1038/ncomms9357) and activation of YAP could be an indirect feedback that is adopted by cells in response to kinase inhibition (DOI: 10.1016/j.trecan.2019.02.010).

4. As the RNA-Seq omic data shows upregulation of YAP-target genes including CYR61, CTGF, Ajuba, Birc5, Myc etc. Have the authors revalidated those TEAD-YAP signature by RT-PCR in supporting cells?

5. Does TRULI also activate TAZ in a similar way through LATS1/2 inhibition? If it's the case, does TAZ play an individual/redundant role upon TRULI treatment? Should include this data in the manuscript or supporting material.

6. Why treatment of confluent cells with TRULI increase Mob1 phosphorylation in MCF10A cells while it showed no effect on pMob1 in HEK293A cells with/without serum starvation?

Minor points:

1. The authors should update recent progress about current YAP activators in the introduction section such as TEAD activators (DOI: 10.1021/acscchembio.9b00786), MST inhibitors (DOI: 10.1126/scitranslmed.aaf2304) etc.

2. There are many MDA-MB cell lines. The exact mammary-gland line should be described.

3. The Z' factor of phenotypic screen assay should be included to evaluate the assay stability.

We thank the reviewers for their insightful and encouraging comments! The experimental and theoretical investigations they motivated have substantially strengthened the manuscript. Please find below our point-by-point response to each comment.

REVIEWER COMMENTS

Reviewer #1:

This is a well-done and well-written study on Hippo signaling in the mouse inner ear and cardiomyocytes, and human Muller glia in retinal organoids. In all three organ systems, regeneration in humans is restricted/ blocked. Thus, any efforts to understanding these blocks is a significant advance in the biomedical field. When the Hippo pathway is in its active state, the transcription factors, Yap/Taz are targeted for degradation and thereby, inhibiting proliferation. When the Mst and Lats proteins are inhibited, the pathway is turned off, allowing Yap/Taz to activate transcription of G1-S and G2-M check point genes. In this collaboration, the authors mechanistically test the Lats inhibitor, TRULI, to stimulate proliferation.

This study is beneficial and of general interest to the biomedical field and I recommend this manuscript for publication if the authors can satisfy the requirements below. There are a few issues with some experiments with sample sizes and the appropriate statistical tests employed.

Major Comments:

1. The experiment in the cochlea described on page 6 (Figure S3) was done on P1 cochleas. Samples size is not noted, unlike in Fig. S2 (utricle). Such proliferative effects

at this stage of development are usually only seen in the apex. What part of the cochlea were these images from? Base, mid, apex? In Fig. S3B, HCs in the TRULI condition appear to be 'blebbing', which can be a sign of toxicity (in the cochlea). Please comment on this and note how many samples exhibit this. There appears to be absolutely no proliferation in the organ of Corti itself. Is the cell cycle inhibitor, p27kip1 modified in any way?

The treatments of the organ of Corti were repeated twice with three individual cochleae each time; the number of experiments is now noted in the Figure S3 legend. Images were acquired at a mid-basal turn and a similar pattern was observed in all the samples. Because an optical section through the supporting cell nuclei is shown in panel B, only the very basal portions of a subset of the inner hair cells can be seen. Blebbing is usually evident at the apical surface of the hair cells and was not observed in either control or TRULI-treated samples (as shown in panel A). Hair cell survival after the treatment was also directly assessed in the Figure 5D, in which the absence of a change in the number of sensory receptors is quantified. Finally, in Figure 4C we demonstrate that pro-apoptotic genes are repressed by TRULI treatment. These data are consistent with no detectable toxicity of the drug in the hair cells and with pro-survival effects of Yap activation in the supporting cells.

P27kip1 was not modified in these experiments; thank you for this interesting point. It is very likely that Hippo and p27Kip1 act in parallel to maintain quiescence in the organ of Corti. We provide the first evidence for the role of Hippo in the development of the organ of Corti in a recently published paper (Gnedeva et al., 2020) that we now cite in the manuscript. We plan on following up on these findings and other differences in the maintenance of a postmitotic state in the vestibular and auditory sensory epithelia—work that we believe to be outside the scope of the current manuscript.

2. On page 7, “Even under treated, serum-fed conditions, phosphor-Yap levels were below those of serum-fed control cells.” What do you mean by ‘under treated’?

‘Under’ was meant to suggest within that condition, treated or serum-fed. This has been changed to “Even in the treated, serum-fed condition, phosphorylated Yap levels were below those of serum-fed control cells”

3. In Figure 2 (page 35),

D- The authors mention the number of cells, but not the number of organs used. Please note, N= number of experiments. The authors should sample across more than 1 utricle under both conditions. Please clarify.

It was n=2 utricles for both conditions; the figure legend has been updated to reflect this.

E- The sample size noted is n=2 and uses a student’s t-test. First, sample sizes n=2 to 3 aren’t sufficient to ensure normal distribution of samples that is required for a t-test. Second, you can’t use a t-test to compare across 3 samples in the way the authors show. You must use an ANOVA.

G- same problem. Can’t use a t-test here. Must use an ANOVA.

Under the reporting summary, “A description of any assumptions or corrections, such as tests of normality and adjustment for multiple comparisons” is applicable and must be satisfied. Although significance may not change with additional samples, it is important that the correct statistical tests are applied.

In both cases E and G, the sample sizes are too small to show normal distribution. In my experience, an $n=5$ could demonstrate this. But normal distribution needs to be determined first.

As requested, ANOVA was performed instead of Student's t-test, and the results were incorporated into the figure legends. These statistical analyses confirmed the significance of the observed changes in both E and G. We additionally note that in the control condition, for which no proliferation of the supporting cells is observed, an increase in N would not be sufficient to demonstrate normal distribution around zero. We believe that the binary nature of the results is rather striking.

4. Inconsistencies in Figure 6 (page 42). In F and G, across how many experiments were these measurements taken? For example, in K, the authors clearly noted that there were 3 experiments.

Across four experiments; the figure legend has been updated to reflect this.

D-G. Since several comparisons are being made from (supposedly) the same data, a correction factor to the p-value must be applied e.g Bonferroni correction.

In the authors' assessment, Bonferroni correction should be applied to an assay in which several tests are performed on the same data with distinct criteria, without a predetermined hypothesis (Armstrong, RA; doi: 10.1111/opo.12131). Here all four, highly correlated measurements are of a single hypothesized phenomenon, mitosis. Moreover, the test is useful in stringently eliminating potential occasional, seemingly significant

p-values when most are insignificant. Here all the tests are significant, and three of the four tests yield $p < 0.0001$. We do not think that this is a warranted change.

5. In the legends on Figure 5 (page 41), I'm not sure why the authors only make the full list of DE genes upon request. Shouldn't this be a downloadable file from GEO?

At the time of the original submission, data deposition was not required. We have since uploaded all the RNA-sequencing data to the NCBI's Gene Expression Omnibus (GSE148528). This accession number is now listed in the appropriate figure legends and the data accessibility statement.

6. I understand that the authors only used an $n=2$ for RNA sequencing; however, in this case secondary validation of a few selected genes by RT-qPCR (or in situ hybridization) is necessary as RNA sequencing is not sensitive enough and can throw out false positives. Especially in this case, when proliferation is affected. More cells than skew quantitative results. In this case, I recommend RT-qPCR.

To confirm our observations, we utilized RNA from three biological samples collected at day 5 and day 10 for RT-qPCR validation. We selected highly conserved downstream targets of Yap-Tead signaling (for day 5) and hair cell genes (for day 10) identified as significantly differentially expressed in the RNA-sequencing analysis. In particular, Ctcf, Cyr61, and Ccnd1 were shown as direct Yap targets in the mammary epithelial cell lines (Zanconato et al., 2015), and, more recently, confirmed by us in the cochlea (Gnedeva et al., 2020). Atoh1, Pou4f3, and Pvalb were selected as a highly-conserved hair cell genes. All six selected genes were confirmed by RT-qPCR as significantly upregulated after treatment with TRULI.

Below we demonstrate the results for your reference. All gene expression was normalized to GAPDH expression in the same sample and to DMSO-treated control levels to compare the relative expression levels.

Because these data do not add to the conclusions of the manuscript, but provide a validation of the RNA-sequencing technique itself, we would prefer not including them to the manuscript. The quality and sensitivity of RNA-sequencing technology have improved greatly over the past decade, making it a highly reliable method for characterization of genome-wide transcriptome (SEQC Consortium, 2016). Although we agree with the reviewer that two biological samples may provide less information than three, the principal outcome of decreasing the number of biological replicates in RNA-sequencing analysis is an increase in the rate of false negatives (Schurch et al., 2016). In other words, using fewer replicates decreases the number of differentially expressed genes that can be identified with the same confidence (FDR), detecting only the most drastic changes. Using two biological samples is generally accepted as outlined in the data-analysis package utilized (DESeq2). Because our purpose is to broadly describe the biological processes

induced or inhibited by TRULI in the supporting cells, as opposed to identifying all the differentially expressed genes and exact fold changes, we believe that the use of either two or three biological samples is appropriate.

Minor Comments:

7. On page 5, “Every plate also included a positive control, sub-confluent cells, and a negative control, densely cultured cells, both exposed to dimethyl sulfoxide (DMSO) at a concentration equivalent to that in which the compounds were applied.”

Just succinctly refer to it as a ‘DMSO vehicle control’. It is understood that you used the same concentration of DMSO.

This modification has been made.

8. TRULI is first mentioned on page 6, but its abbreviation is described very late on page 8, as the “The Rockefeller University Lats inhibitor”. This should be mentioned earlier on page 6.

At the first mention of the name, we now note that "For the sake of brevity—and as justified below—we term this substance 'TRULI' for 'The Rockefeller University Lats Inhibitor.' "

9. Figure S6, In the figure title/ legend please clarify that these are Utricles.

This modification has been made.

10. On page 11, "...we treated P21 utricles with TRULI and virally transfected the cultures with Atoh1-RFP upon drug withdrawal." The way this reads, Atoh1-RFP is a reporter for Atoh1. If I am not mistaken, the Methods section describes it as an 'Atoh1-RFP expression construct'. If the authors intended only to observe the capacity for HC differentiation upon TRULI treatment, please explain the rationale for using an expression construct rather than a reporter construct.

Atoh1-RFP is, in fact, an Atoh1 expression vector harboring RFP reporter to label transduced cells. We note earlier in the manuscript that despite seeing transcriptional upregulation of a large number of hair cell genes in the TRULI withdrawal condition, Pou4f3+/EdU+ cells are not detected by antibody labeling. It is important to add that, although hair cell-specific genes are upregulated upon TRULI treatment as compared to the controls, these genes are still expressed at the relatively low levels. The FPKM values detected in hair cells (Menendez et al., 2020), are significantly higher than that after TRULI-induced cell cycle reentry. This may also suggest that the hair cell-specific RNAs, upregulated upon TRULI treatment, are not being translated into the proteins.

To test whether the progeny of supporting cell proliferation, that we observe after TRULI treatment, can be directed to differentiate into hair cells, we transduce these daughter cells with the Ad5- Atoh1-RFP viral vector, which is highly tropic towards supporting cells (Gnedeva and Hudspeth, 2015). After Atoh1 is overexpressed, many Pou4f3+/EdU+ cells progeny of supporting cells can be detected, which strongly suggests that these cells retain the capacity to differentiate towards the sensory receptors after undergoing a mitosis.

11. On page 13, second paragraph, 'cochlea' is mis-spelled as 'cocTiktaalikhlea'.

This modification has been made.

Reviewer #2:

The authors report the identification of a small molecule that inhibit the Lats kinases, which are key tumor suppressors in the Hippo pathway. This signaling pathway is highly conserved and an essential regulator of development and disease. Pharma and academia are pursuing efforts to target it for therapeutic benefit, predominantly in the context of cancer (mostly Yap, TAZ, TEAD inhibitors) and regenerative medicine. There has been some progress on the latter front, but this has centred on the upstream Hippo kinase (human MST1 and MST2). This study is original by virtue of the fact that it centres on the key downstream kinases, Lats1 and 2, which are more potent (based on genetic loss of function studies in flies and more recently mammals).

The authors use an elegant cell density screen to identify a handful of compounds that reinstate nuclear Yap under confluent conditions and focus on TRULI. They subsequently show that this is an ATP-competitive inhibitor of Lats1 and 2. They convincingly show using RNA-seq that Yap hyperactivation is a major downstream effect of the drug. They also show that it has promising regenerative capacity in different tissues, most notably the utricle. TRULI induces cell cycle re-entry but also plasticity and this enable newly generated cells to adopt specific fates, which is important for tissue repair. TRULI should be a useful tool compound for Hippo studies and is a promising candidate to be developed for potential use in regenerative medicine, especially because it should be more potent than MST1/2 inhibitors for Yap activation. Its study will also help to determine whether short term Hippo pathway disruption is safe and does not induce unwanted effects, like YAAP-driven oncogenesis.

This is a high-quality study and should be published in Nature Comms. I have only one suggested experiment (which is essential) and minor comments in order to improve the paper.

The studies in the utricle and retinal organoids report that TRULI induces proliferation, based on EdU positivity. The authors have not actually shown that cell number increases in response to TRULI, just that DNA replication is stimulated. Cell number and mitosis (e.g. with phospho-histone H3) should be assessed in order to more rigorously test this key assertion. I note that they have assessed mitosis in cardiomyocytes but cell number should also be tested here and mitosis and cell number should be assessed in the other settings.

We believe that the fundamental question is whether cells progress through the cell cycle or arrest prematurely. Unfortunately, counting the total number of supporting cells in the utricle—on the order of 6000 cells per organ—would be technically challenging, extremely laborious, and imprecise. The addition of a few hundred new, EdU-positive supporting cells would therefore be statistically hard to demonstrate. This approach would require impeccable labeling and volumetric three-dimensional imaging and analysis; to detect a significant difference of that size would require significantly more samples.

The heterogeneous nature of retinal organoids makes them inappropriate for demonstrating increases in cell numbers: organoids are composed of several cell types and are largely variable in size. One can neither compare within a single treated sample both before and after, nor compare absolute numbers between organoids.

As requested, we assessed for pH3 and AurkB, markers of mitosis, in the utricle after 5 days of TRULI treatment. As quantified below and in the attached figure, we found a

large number of supporting cells that were positive for both, EdU and one of these G2/M markers. Two 60X fields of view apiece were examined from three utricles. We quantified the total number of EdU-positive cells and EdU-positive cells that were also AurkB or pH3 positive; we also calculated the corresponding percentages.

We note, in addition, that our RNA-seq data from the utricle demonstrate active G2/M programs—including cyclins, as demonstrated below—indicating that the supporting cells progress through the S phase of the cell cycle. This result buttresses our other evidence, described next, that the cells in fact progress through mitosis.

Finally, the experiments with drug withdrawal demonstrate that supporting cells labeled with EdU in the first 5 days survive in culture for an additional 5 days and are capable of differentiation towards a hair cell fate upon Atoh-RFP transduction. These data strongly suggest that these progeny of supporting cells are not arrested at the G2/M checkpoint.

We believe taken together this evidence demonstrates that Lats inhibition via TRULI promotes re-entry into, and progression through, the cell cycle.

Some comments on the text:

- Despite the fact that the Hippo pathway is commonly described as being in active or inactive states and Yap and TAZ to be sequestered in the cytoplasm and degraded in response to Hippo, live imaging studies in cultured cells and flies have now shown these assumptions to be largely wrong. In fact, the authors' own data (Figure 1A) reveals the limitations of this broad statement, with respect to Hippo signaling and Yap degradation. These assumptions should no longer be perpetuated in the literature.

*We agree that this is an oversimplification, and have amended the third paragraph of the introduction to say "When proliferation is unwarranted, Mst1 and Mst2 phosphorylate Lats1 and Lats2; these proteins in turn phosphorylate the transcriptional co-activator Yap and its homolog Taz, which as a result are **predominantly** sequestered and degraded in the cytoplasm. When the pathway is inactive, Yap **accumulates** in the nucleus, interacts with transcription factors of the Tead family, and initiates cell division"*

-All statistical information (error tests, n numbers, etc.) should be provided in the figure legends. This is not always provided.

These have been provided.

-Legends for Figures 2, 4, 5 and 6 need more descriptive titles. Non-specialists won't know what "supporting cells" are. Add the organ name.

We have corrected Figure titles to include names of the organ systems used.

Reviewer #3:

Comments to the authors:

Herein, Kastan et al. reported the discovery of non-toxic, reversible Lats1/2 kinase inhibitor namely TRULI through high throughput phenotypic screen, which is a very important and inviting topic in regeneration medicine. TRULI promoted nuclear Yap translocation and evoked supporting-cell proliferation. Direct binding to Lats1/2, ATP competitive MOA and global kinome selectivity were demonstrated in in vitro kinase assays. The upregulation of cell cycle related genes and Yap target genes triggered by compound was evaluated by RNA-Seq. Additionally, TRULI could induce supporting-cell differentiation and stimulate the proliferation of cardiomyocytes and Müller glial cells. Overall, it's an interesting and solid study. However, several issues need to be addressed to increase the confidence in the specificity.

Major points:

1. TRULI shows Lats1/2 inhibition with IC₅₀ value of 0.2 nM while cellular potency differs by three orders of magnitude (EC₅₀ = 510 nM). Any explanation for this discrepancy? Some phenotypic experiments were conducted in 10 μM, which is significantly higher than in vitro biochemical potency. Is that due to poor cell permeability or mM-level cellular ATP concentration, which positively shifted IC₅₀ of ATP-competitive TRULI in cells (indicated from Figure 3E)?

The difference in potencies is to be expected as one progresses from in vitro kinase assays (IVKAs) to in-cell potency to explant assays. As the reviewer suggests, the primary explanation is that the ATP concentrations are orders of magnitude apart, which directly affects the apparent potency of an ATP-competitive compound. Other factors might include the contribution of negative feedback within the cell or non-specific binding

to serum proteins, which could decrease the free fraction of a small molecule. The ATP concentration inside a cell is close to 5 mM, whereas the IVKAs were conducted with 10 mM of ATP, a value originally chosen to allow the detection of less potent compounds. In fact, when we perform the IVKA with 2 mM ATP, the IC₅₀ lies in range of a few hundred nanomolar, consistent with our in-cell EC₅₀ assay. In keeping with the standard practice in the application of small molecules for biological systems, experiments are performed at concentrations severalfold above the in-cell IC₅₀ values. IC₅₀ is the concentration at which the compound is 50 % active, so we work at concentrations about twenty-fold as great. The IVKA value is most useful for comparing among other compounds within a particular assay system and has little broadly interpretable value. The in-cell assay is much more appropriate for a sense of potency within cells. Effective concentrations must ultimately be demonstrated in context-specific settings.

2. Also concerning the selectivity issue, it would be better to show Lats1/2 KO result or introduce resistant mutation based on docking model to make Lats1/2 refractory to inhibitor to demonstrate the observed phenotype is due to direct Lats1/2 inhibition. Alternatively, or in addition, it would be good to show basic structure-activity relationships for this compound. This would help address the question of whether the activity is primarily due to LAT1/2 inhibition or the consequence of polypharmacology. So showing correlation between biochemical LATs1/2 inhibition and cellular activity would lend support to the notion that the activity is indeed 'on-target'.

Rudolph et al. (DOI: <https://doi.org/10.1523/JNEUROSCI.0306-20.2020>) recently published Lats1/2 knockout data for the utricle, and their results were consistent with ours: loss of Lats activity is itself sufficient to drive proliferation, and does not require invocation of off-target mechanisms. Unfortunately, this system would not allow the further demonstration of specificity: as the paper showed, Lats1/2 knockout induces robust

proliferation, and thus to treat double-knockout specimens would not allow assessment whether TRULI treatment is specific to Lats1/2 inhibition: we would be comparing two highly proliferative conditions. Moreover, we used the Yap-knockout experiment to show that the proliferative effects in that context are largely—if not completely—driven by Yap. This result demonstrates that the mechanism of action is the activation of Yap, which was the original goal of the endeavor. Other experiments in the manuscript also strongly suggest direct inhibition of Lats as the primary mechanism of Yap activation by TRULI. The in vitro experiments with HEK293A and MCF10a cells (Figure 3) showed that Lats is successfully put into an “active” state by the cell, yet Yap remains unmodified. Even if TRULI had effects on relevant upstream targets such as Mst1/2 or Map4k4 (regarding the next comment), those effects being pre-dominant is not supported by our work. There are no published pathways that could bypass Lats to inhibit Yap, and it therefore seems unavoidable to infer that TRULI’s effect are at least predominantly mediated by Lats inhibition and Yap activation. We also have data indicating that Yap activation occurs in minutes in vitro in HEK293A cells, suggesting immediate activation and not feedback mechanisms.

We could obtain a definitive proof if we could identify an amino acid within the ATP pocket of Lats1/2 that permits the binding of ATP but not TRULI. However, it is important to note there is no crystal structure for Lats1 or Lats2, and obtaining that information would likely merit a separate manuscript. Moreover, it is likely that mutations that disturb TRULI binding would also effect ATP binding, rendering the experiment moot. Finally, to control for normal levels of Lats1/2 and to replace endogenous protein, it would be necessary to generate a knock-in mouse for this single demonstration.

We do in fact have a suite of compounds that demonstrate the suggested structure-activity relationship, but that material is subject to a patent application, and we believe

our work included within the manuscript in itself sufficient to indicate that the mechanism of Yap activation by TRULI is through direct Lats inhibition. Although we are working on several follow-up studies, we believe that the congeners are beyond the scope of this paper.

3. The authors tested kinome selectivity (314 kinases) via in vitro binding assays. 34 kinases bound more strongly than Lats1/2, although it's acceptable considering it's an unoptimized hit. Would be interesting to see cell-based selectivity on this. ActivX Biosciences provide KiNativ™ that could be considered for in situ kinase profiling and on-target validation in cell models. Also, many kinases such as MAP4K would have crosstalk with Hippo signaling cascade (DOI: 10.1038/ncomms9357) and activation of Yap could be an indirect feedback that is adopted by cells in response to kinase inhibition (DOI: 10.1016/j.trecan.2019.02.010).

As noted in the comments above, we believe the data identify Lats inhibition as the key effect of TRULI. As the reviewer notes, a starting compound likely inhibits off-target effects that would be of concern in potential therapeutic applications. As a tool compound, however, TRULI is relatively clean with a clear mechanism of action. We note that—as mentioned in our manuscript—there are clinically approved drugs with similar binding scores, and reiterate that binding does not directly correlate with functional inhibition, as demonstrated by some of the functional IC₅₀ determinations on off-targets from the list. To this point, and the reviewer's point about MAP4K4, please see the comments in the answer above, and also note that MAP4K4 sits lower on the list than ROCK1, for which we did a functional kinase assay and found the inhibition quite weak. Our in vitro work first identified Lats as a potential target because of the clear continued phosphorylation of Lats and prevention of Yap phosphorylation. In two distinct biological systems, that evidence supports the inference that direct Lats inhibition is the mechanism of TRULI-

mediated Yap activation. And finally, as the reviewer notes, this is a starting compound; our goal will be to develop superior compounds in terms of potency and specificity.

4. As the RNA-Seq omic data shows upregulation of Yap-target genes including CYR61, CTGF, Ajuba, Birc5, Myc etc. Have the authors revalidated those TEAD-Yap signature by RT-PCR in supporting cells?

We now provide the RT-qPCR validation of these data in our response to Reviewer #1 (please see above).

5. Does TRULI also activate TAZ in a similar way through Lats1/2 inhibition? If it's the case, does TAZ play an individual/redundant role upon TRULI treatment? Should include this data in the manuscript or supporting material.

Although the answer to this question is likely-context dependent, our prediction would be that any context in which Lats and Taz are signaling, Taz would also be effected by TRULI treatment. One area in which this relationship has been characterized is in serum-starved HEK293 cells (Plouffe et al., doi: 10.1074/jbc.RA118.002715). The authors showed that in contrast to Yap, Taz is primarily degraded in response to Lats activity, and if Lats is knocked out, Taz levels are elevated in basal conditions and fail to decrease in a serum-starved condition. We would accordingly predict that serum starvation in the presence of TRULI would prevent degradation of Taz. To test our prediction, we conducted the serum-starvation treatment of HEK cells used in our manuscript. We are attaching a figure in which controls demonstrate that serum starvation decreases Taz levels, but when the cells are pre-treated with TRULI, basal levels of Taz are elevated, and the cells no longer robustly degrade Taz. This fits nicely in our paradigm as theory would suggest.

As we are tight for space in our manuscript, we are happy to share the data, but would prefer not include them on the ground that they do not alter our conclusions. If the reviewer would strongly prefer we include this, we will do so if the Editor permits additional space.

6. Why treatment of confluent cells with TRULI increase Mob1 phosphorylation in MCF10A cells while it showed no effect on pMob1 in HEK293A cells with/without serum starvation?

The stimulation of Lats is distinct in the two contexts: in MCF10A cells in terms of density, mediated by the canonical pathway including Mst1/2 and Mob1; and in HEK293A cells as elicited by serum starvation, which elicits AMPK-mediated activation of Lats, bypassing Mst1/2 and Mob1 (Mo et al.; doi: [10.1038/ncb3111](https://doi.org/10.1038/ncb3111)). Please note that the HEK293 assay is done after 1 hour of starving and treatment, whereas the MCF10a cells are treated for 24 hours; this could also contribute to the extent and type of negative feedback.

Minor points:

1. The authors should update recent progress about current Yap activators in the introduction section such as TEAD activators (DOI: 10.1021/acscchembio.9b00786), MST inhibitors (DOI: 10.1126/scitranslmed.aaf2304) etc.

We have added the first reference, thank you.

Regarding the second reference, please see Figure S2 where we directly compared the effects of TRULI and XMU on utricular supporting cells. These results were also deliberated in the Discussion.

2. There are many MDA-MB cell lines. The exact mammary-gland line should be described.

Thank you for pointing this out: MDA-MB-231 cells were utilized. This has been corrected in the text.

3. The Z' factor of phenotypic screen assay should be included to evaluate the assay stability.

The Z-factor, which is commonly used to evaluate an assay's reproducibility, reflects consistency of the difference between the negative and positive controls utilized in a screen. Because a true positive control—a confluent culture treated with a known inducer of Yap nuclear translocation—did not exist at the time, the Z-factor was not assessed. To demonstrate our assay's reproducibility, we instead included Figure S1, in which ten 384-well screening plates and over 3500 compounds are demonstrated. As one can appreciate, a minimal variation in Yap nuclear translocation and cell density is seen in the

DMSO-treated confluent cultures—the only control used for determining hits, as described in the Materials and Methods.

REVIEWER COMMENTS

Reviewer #1 (Remarks to the Author):

The changes applied to the manuscript and addressing reviewer comments are satisfactory.

I recommend this manuscript for publication.

Reviewer #2 (Remarks to the Author):

The authors have attended to some suggestions with new experiments and rebutted the need to do others. As such the manuscript is improved.

One final thing they still need to clarify is the introduction. They have amended it as such: "We agree that this is an oversimplification, and have amended the third paragraph of the introduction to say "When proliferation is unwarranted, Mst1 and Mst2 phosphorylate Lats1 and Lats2; these proteins in turn phosphorylate the transcriptional co-activator Yap and its homolog Taz, which as a result are predominantly sequestered and degraded in the cytoplasm. When the pathway is inactive, Yap accumulates in the nucleus, interacts with transcription factors of the Tead family, and initiates cell division"

However, this is still incorrect. There is no evidence in any of the hundreds of papers on the Hippo pathway that it exists in on and off states and that it is modulated when "proliferation is unwarranted" or any other signal for that matter. Further, live imaging studies have shown that the Hippo pathway does NOT predominantly regulate YAP and TAZ and Yorkie by cytoplasmic sequestration and degradation. Rather, the pathway tunes the rate of flux of these proteins between the nucleus and cytoplasm. Cytoplasmic of YAP and TAZ and sequestration plays a very minor role on their subcellular localization and the overall impact of degradation is still not well defined.

This should be amended - there are many recent papers (primary and review) on this topic where the relevant facts can be assessed.

Reviewer #3 (Remarks to the Author):

The authors have been insufficiently responsive to my reviews. The evidence provided that this compound is functioning as a selective Lats inhibitor is weak. The kinase selectivity of this compound is poor and there is insufficient evidence provided that the pharmacology is derived from Lats inhibition. Inhibitor resistant rescue (best) or showing correlation between Lats inhibitor biochemical inhibition and cellular pharmacology is needed. Otherwise the authors should report this as an interesting phenotypic screening hit and greatly reduce the claims to having found a selective Lats inhibitor. The authors can file a patent on their new compounds and show the correlation between biochemistry and cellular pharmacology, they don't need to disclose their chemical structures for the review process.

I also wasn't able to locate the chemical characterization data for the compound (^1H NMR, LC-MS, etc).

Once again, we thank the reviewers for their time and comments. Please find below our point-by-point response to each of the final issues raised by two of the reviewers.

REVIEWER COMMENTS

Reviewer #1

The changes applied to the manuscript and addressing reviewer comments are satisfactory.

I recommend this manuscript for publication.

Reviewer #2

The authors have attended to some suggestions with new experiments and rebutted the need to do others. As such the manuscript is improved.

One final thing they still need to clarify is the introduction. They have amended it as such: "We agree that this is an oversimplification, and have amended the third paragraph of the introduction to say 'When proliferation is unwarranted, Mst1 and Mst2 phosphorylate Lats1 and Lats2; these proteins in turn phosphorylate the transcriptional co-activator Yap and its homolog Taz, which as a result are predominantly sequestered and degraded in the cytoplasm. When the pathway is inactive, Yap accumulates in the nucleus, interacts with transcription factors of the Tead family, and initiates cell division' "

However, this is still incorrect. There is no evidence in any of the hundreds of papers on the Hippo pathway that it exists in on and off states and that it is modulated when “proliferation is unwarranted” or any other signal for that matter. Further, live imaging studies have shown that the Hippo pathway does NOT predominantly regulate YAP and TAZ and Yorkie by cytoplasmic sequestration and degradation. Rather, the pathway tunes the rate of flux of these proteins between the nucleus and cytoplasm. Cytoplasmic of YAP and TAZ and sequestration plays a very minor role on their subcellular localization and the overall impact of degradation is still not well defined.

This should be amended - there are many recent papers (primary and review) on this topic where the relevant facts can be assessed.

We have implemented this suggestion by adding a recent review of Yap flux. Moreover, we have amended the Introduction on page 4 to state "The canonical Hippo pathway is a highly conserved signal-transduction cascade that comprises two pairs of core kinases. When activated by upstream signals, Mst1 and Mst2 phosphorylate Lats1 and Lats2; these proteins in turn phosphorylate the transcriptional co-activator Yap and its homolog Taz, adjusting the flux of these proteins so as to favor cytoplasmic localization. When the phosphorylation cascade is inactive, Yap flux into the nucleus is enhanced, leading to interaction with transcription factors of the Tead family and the initiation of cell division^{12,13}.

Reviewer #3

The authors have been insufficiently responsive to my reviews. The evidence provided that this compound is functioning as a selective Lats inhibitor is weak. The kinase selectivity of this compound is poor and there is insufficient evidence provided that the pharmacology is derived from Lats inhibition. Inhibitor resistant rescue (best) or showing correlation between Lats inhibitor biochemical inhibition and cellular pharmacology is needed. Otherwise the authors should report this as an interesting phenotypic screening hit and greatly reduce the claims to having found a selective Lats inhibitor. The authors can file a patent on their new compounds and show the correlation between biochemistry and cellular pharmacology, they don't need to disclose their chemical structures for the review process.

We do not claim that we have identified a perfectly selective inhibitor, but in vitro kinase assay demonstrates that TRULI directly interferes with Lats kinase activity. Further, kinome-wide binding panel demonstrate that it has a good selectivity score, comparable to those of some medications currently in clinical use. Moreover, we also showed that some of the top enzymes from the predicted list are not actually inhibited by TRULI. Finally, we demonstrated in the two cell lines that the Hippo pathway is active up to—and inclusive of—Lats activation, yet Yap remains unmodified. So far as we are aware, the literature regarding the Hippo pathway offers no other explanation of how Yap would remain unphosphorylated in the presence of activated Lats, if not through the action of a Lats inhibitor. Even though iterative rounds of medicinal chemistry would improve selectivity and potency of the compound, these evidence demonstrate that TRULI is an inhibitor of Lats kinases, the main point of our manuscript.

Accepting that the evidence for Lats inhibition, although strong, is nevertheless circumstantial, we have modified the text at several sites. On page 9, we note that "Before considering therapeutic uses of TRULI or a related compound, it will be necessary to explore other potential off-target kinases, particularly in a tissue-specific context. TRULI is a lead compound, the first "hit" in a small-molecule screen, and there is no doubt that any clinical uses of the associated family of substances would require extensive medicinal chemistry to increase potency, evaluate and presumably diminish cross-reactivity, and achieve a form suitable for dosing. Our data nonetheless suggest that TRULI is a potent, direct, and relatively selective inhibitor of Lats1 and Lats2." On page 14, we now state "As a putative inhibitor of Lats kinases, TRULI also offers the ability to investigate these non-Hippo functions, and it would be prudent to understand the implications of Lats inhibition on these pathways prior to therapeutic endeavors." And in the final paragraph of the Discussion on page 15, we state that "Although our evidence suggests that TRULI is a potent, non-toxic, and reversible inhibitor of Lats kinases, a definitive demonstration will require crystallographic confirmation of the compound's binding site."

I also wasn't able to locate the chemical characterization data for the compound (1H NMR, LC-MS, etc).

Below we provide the requested chemical characterization data for the compound. Please note that TRULI has a numerical name of RU5757.

Nuclear Magnetic Resonance Spectrogram –TRULI (RU5757).

Liquid Chromatography–Mass Spectrogram –TRULI (RU5757).